# Modelling of onchocerciasis-associated skin and ocular disease and the impact of ivermectin treatment
Matthew A. Dixon [1,8] ✉, Aditya Ramani [1,2,8], Martin Walker [1,2], Jacob N. Stapley[1], Michele E. Murdoch[3], Ian E. Murdoch[4], Gladys A. Ozoh[5], Jonathan F. Mosser[6,7] & Maria-Gloria Basáñez [1] ✉

## Abstract

**Background** Despite decades of control interventions in sub-Saharan Africa, morbidity associated with *Onchocerca volvulus* infection still exerts a substantial burden of disease, arising from cutaneous, ocular and neurological manifestations.
**Methods** We developed and integrated a morbidity sub-model into our previously published individual-based, stochastic transmission model, EPIONCHO-IBM, including both reversible (severe itch, reactive skin disease (RSD)), and irreversible (skin atrophy, depigmentation, hanging groin) cutaneous sequelae, and eye disease (blindness, visual impairment). We modelled the relationship between onchocerciasis skin disease (OSD) and infection prevalence using pre-intervention data from northern Nigeria, and between onchocerciasis ocular disease (OOD) and infection intensity using data from the Onchocerciasis Control Programme in West Africa. We simulated the impact of ivermectin mass drug administration (MDA) upon OSD and OOD using data from Cameroon, Central African Republic, Nigeria, Sudan and Uganda.
**Results** Modelled age-specific OSD and OOD prevalence at baseline align well with reported prevalence estimates across the simulated range of endemicity levels but underestimate irreversible OSD in older age groups. Under MDA, we capture trends in infection prevalence, severe itch and irreversible OSD but underestimate reductions in RSD and blindness prevalence.
**Conclusions** Integrating morbidity outcomes into transmission dynamics modelling will help improve estimates of onchocerciasis disease burden and inform the effectiveness and cost-effectiveness of current and alternative interventions.

## Plain Language Summary

Onchocerciasis (river blindness) is a neglected tropical disease caused by a parasitic worm. Despite decades of control programmes, it still affects millions of people across Africa, leading to skin and eye disease as well as epilepsy. In this study, we incorporated skin and eye clinical manifestations as outcomes of our EPIONCHO-IBM transmission model, which uses mathematical equations to project how mass treatment with the anti-parasitic drug ivermectin affects the prevalence of these conditions over time. The model captured many of the patterns seen in real-world data, but underestimated improvements in some skin conditions and vision loss following long-term treatment. Including disease outcomes in transmission models can enhance estimates of the true impact of river blindness and help guide effective interventions for its control and elimination.

Human onchocerciasis (river blindness), caused by the filarial nematode *Onchocerca volvulus*, remains endemic (as of 2025) across 28 sub-Saharan Africa (SSA) countries, despite decades of interventions[1]. The majority of cases (99%) and disease burden are in SSA, with the 2017 Global Burden of Disease (GBD) Study estimating that 14.6 million of infected people had

skin disease and 1.15 million had vision loss[2]. Two regional programmes, the Onchocerciasis Control Programme in West Africa (OCP) (1975–2002), covering 11 countries[3] and the African Programme for Onchocerciasis Control (APOC, 1995–2015), covering another 20 countries[4], have implemented large-scale interventions in SSA. The main strategies have focused

[1]MRC Centre for Global Infectious Disease Analysis and London Centre for Neglected Tropical Disease Research, Department of Infectious Disease Epidemiology, School of Public Health, Imperial College London, London, UK. [2]Department of Pathobiology and Population Sciences, Royal Veterinary College, Hatfield, Hertfordshire, UK. [3]Department of Dermatology, West Herts Teaching Hospitals NHS Trust, Watford, Hertfordshire, UK. [4]International Centre for Eye Health, Institute of Ophthalmology, London, UK. [5]Dermatology Division, University of Nigeria Teaching Hospital, Ituku Ozala, Enugu State, Nigeria. [6]Institute for Health Metrics and Evaluation, Hans Rosling Center for Population Health, Seattle, WA, USA. [7]Department of Global Health, School of Public Health, and Seattle Children's Hospital, University of Washington, Seattle, WA, USA. [8]These authors contributed equally: Matthew A. Dixon, Aditya Ramani. ✉e-mail: m.dixon15@imperial.ac.uk; m.basanez@imperial.ac.uk

on vector control, particularly in West Africa, to tackle the *Simulium* blackfly vector[3] and mass drug administration (MDA) with ivermectin to tackle infection in humans[3,4]. After the closure of these programmes, endemic countries have continued implementing interventions supported by the Expanded Special Project for Elimination of Neglected Tropical Diseases (ESPEN)[5,6]. Despite these major initiatives being prolonged (nearly three decades for OCP, and two decades for APOC), a recent systematic review found that elimination of transmission has only been reported in 8.5% of SSA foci, in 7 countries[7]. Elimination (interruption) of transmission is the goal of the World Health Organization 2021–2030 Neglected Tropical Diseases (NTD) Roadmap, which proposes that by 2030, 12 onchocerciasis-endemic countries (approximately a third) be verified for elimination[8]. In SSA, and as of 2025, only Niger (a former OCP country) has been verified by WHO[9,10].

The 2021 GBD Study attributed 1.26 (95% uncertainty interval (95% UI) = 0.75–1.90) million disability-adjusted life-years (DALYs) to onchocerciasis, with 20 (95% UI = 18–22) million people infected[11]. In 2024, it was estimated that at least 248 million people in SSA required MDA, not including areas where transmission status is still unknown[6,12]. Ocular and skin morbidity is well recognised[13,14], and increasing evidence indicates an association between onchocerciasis and epilepsy (onchocerciasis-associated epilepsy, OAE)[15]. Despite decades of ivermectin MDA, onchocerciasis clinical manifestations persist in some foci[16].

Onchocerciasis skin disease (OSD) manifests as a range of conditions, including severe itch (capable of detrimentally affecting work and sleep patterns), and reactive skin disease (RSD), which comprises acute papular onchodermatitis (APOD), chronic papular onchodermatitis (CPOD) and lichenified onchodermatitis (LOD)[17]. Skin atrophy (ATR), depigmentation (DPM), and hanging groin (HG) comprise the irreversible OSD sequelae[13]. Onchocerciasis ocular disease (OOD) ensues from the migration of the parasite's progeny stages (microfilariae) to ocular tissue and subsequent damage, upon their death, caused by immune responses to their somatic and endosymbiont (*Wolbachia* bacterium) antigens[18,19]. OOD encompasses a range of clinical manifestations, including those in the anterior chamber of the eye (punctate and sclerosing keratitis; chronic anterior uveitis), and posterior segment (chorioretinitis; optic nerve atrophy), ultimately resulting in loss of visual acuity and blindness[14,20]. A modelling study estimated that prior to MDA in APOC countries, approximately 17.5 million people had some form of OSD or OOD (contributing to 2.5 million DALYs)[21]. Projections from the same study estimated that approximately 4.2 million cases of OSD or OOD would remain by 2030, resulting in 0.7 million DALYs, despite continued MDA[21].

We had previously used a deterministic version of our EPIONCHO transmission model to generate projections of the impact of ivermectin MDA on the prevalence of OSD (severe itch) linked to adult female worm prevalence, and OOD (blindness and visual impairment) linked to microfilarial infection intensity, and estimated pre-intervention disease burden[22]. Using the stochastic, individual-based version of the model, EPIONCHO-IBM[23], we had integrated a published dose-response relationship between microfilarial load in childhood and the probability of developing epilepsy later in life[24] to simulate OAE and the effect of ivermectin MDA on its prevalence and incidence[25]. With a renewed interest from endemic countries, implementation partners, funders and donors in assessing the effectiveness and cost-effectiveness of future intervention strategies to reduce the disease burden of onchocerciasis, this paper presents a modelling approach that incorporates OSD and OOD into EPIONCHO-IBM and projects the impact of ivermectin MDA on their prevalence. Compared with collated data, the updated model captures OSD and OOD age–prevalence patterns and reproduces observed reductions under MDA for severe itch and irreversible OSD.

## Methods

We used individuals' microfilaria (mf)-positivity status to calculate the probabilities of developing OSD and adapted our previous approach[25] for linking an individual's microfilarial count to the probability of developing OOD. Supplementary Fig. 1 presents the probabilities of developing OSD as a function of microfilarial load (not used henceforth because, excepting ATR, the curves were essentially uninformative).

## Epidemiological datasets to derive probabilities of developing onchocerciasis clinical sequelae

**Onchocerciasis skin disease (OSD).** Parameter estimation for OSD used baseline, pre-control cross-sectional data collected in Kaduna State, northern Nigeria (savannah area) in 1988–1989, prior to commencement of ivermectin treatment[26,27]. A total of 6643 individuals aged ≥5 years underwent skin-snip microscopy and clinical examination, across 34 villages where microfilarial prevalence in those aged ≥20 years exceeded 30%[26,27]. Questionnaires were used to assess severe itch. Skin examinations were conducted for RSD (APOD, CPOD, LOD), ATR (in those under 50 years of age), DPM, and HG. Two control non-endemic communities (1342 individuals) were selected where microfilarial prevalence did not exceed 0.3% in those aged ≥5 years to assess background morbidity. We differentiate OSD sequelae into reversible (severe itch and RSD), where an individual can revert to being sequela-negative, and irreversible (ATR, DPM, HG), where an individual remains sequela-positive for the remainder of their life. Proportions for each reversible ($P_{RS}$) or irreversible ($P_{IS}$) OSD sequela were calculated as the proportion of individuals presenting with each sequela among those identified as mf-positive, after subtracting the background proportion of each sequela in the two control communities[26]. For RSD, we considered an individual to be RSD-positive if they had any APOD, CPOD or LOD (some individuals had more than one of these conditions). Supplementary Fig. 2 presents age-prevalence profiles for each condition separately. We assumed that individuals aged <2 years would not develop OSD sequelae. Daily probabilities for the reversible (severe itch and RSD) sequelae were the same as the proportions ($P_{RS}$), calculated as described above. Daily probabilities for irreversible (ATR, DPM, HG) sequelae were calculated using Eq. (1),

$$Pr_{daily_i} = 1 - (1 - P_{IS})^{1/[365 \times (\bar{a} - 2)]} \quad (1)$$

where $Pr_{daily_i}$ is the daily probability of developing each sequela for individual $i$, $P_{IS}$ is the proportion of individuals presenting with the irreversible sequela among mf-positives calculated as described above, and $\bar{a}$ is the average age (in years) of the total examined population. Table 1 presents the (daily) probabilities for each sequela.

## Onchocerciasis ocular disease (OOD)

Blindness due to onchocerciasis was parameterised based on a relationship between microfilarial count (per skin snip) and blindness incidence derived from a longitudinal cohort of 297,756 individuals in 2315 (savannah) villages followed up during the OCP[28]. Individuals were recorded as blind if they had a visual acuity of <3/60 and were unable to count fingers at a 1-m distance with or without perception of light[28]. We used the following log-linear relationship between the probability of developing blindness and microfilarial count,

$$Pr_{blindness_i} = Pr_{background} \times \exp(\gamma_1 \times mf\ count_i) \quad (2)$$

where $Pr_{blindness_i}$ is the probability of developing blindness due to *O. volvulus* infection of individual $i$; $Pr_{background}$ is the background probability of blindness in the OCP area (=0.003, given by a blindness incidence of 300 per 100,000 person-years at 0 microfilarial count); $mf\ count_i$ is the detectable microfilarial count (per skin snip) of individual $i$, and $\gamma_1 = 0.99 \times 10^{-2}$ for 1971–1987 (prior to ivermectin MDA), accounting for a 2-year lag between microfilarial count and blindness onset[28] (Fig. 2). The prevalence of visual impairment was calculated by multiplying the modelled prevalence of blindness by a factor ranging from 0.5 to 1.78 (0.5, 1.0, 1.78), the latter as reported for 338 OCP villages[29] and previously used[22].

**Table 1 | Probabilities of developing onchocerciasis skin disease (OSD) and ocular disease (OOD)**

| Sequelae | Probability | Relationship between probability and individual's *O. volvulus* infection |
|---|---|---|
| Severe itch | 0.164 day$^{-1}$ | Being microfilaria- (mf-) positive |
| Reactive skin disease (RSD) | 0.042 day$^{-1}$ | Being microfilaria (mf-) positive |
| Atrophy (ATR) | [a]4.86 × 10$^{-6}$ day$^{-1}$ | Being microfilaria (mf-) positive |
| Depigmentation (DPM) | [a]7.30 × 10$^{-6}$ day$^{-1}$ | Being microfilaria- (mf-) positive |
| Hanging groin (HG) | [a]2.18 × 10$^{-6}$ day$^{-1}$ | Being microfilaria- (mf-) positive |
| Blindness | Eq. (2), day$^{-1}$ | Individual's microfilarial count |

The proportions of mf-positive individuals presenting with each OSD sequelae in the dataset after subtracting background morbidity[26] were: 0.164 (severe itch); 0.042 (RSD); 0.036 (ATR); 0.059 (DPM), and 0.018 (HG).
[a]Calculated according to Eq. (1).

## EPIONCHO-IBM

We used our stochastic, individual-based model EPIONCHO-IBM, parameterised for savannah onchocerciasis[23]. The model tracks, in a closed population (of 2000 individuals for this work), the number of adult (male and female) worms and skin microfilariae in humans, and infective (L3) larvae in the vector population, with individual exposure to blackfly bites depending on their age, sex, and a factor drawn from a gamma distribution with scale and shape parameter, $k_E$. Parasite population regulation is assumed to be density-dependent and to take place within humans and vectors. The dynamics of infection in the flies (L1, L2, L3 larvae) are modelled deterministically. For each value of $k_E$ (0.2–0.4), there is an associated set of density dependence parameters describing how parasite establishment within humans is determined by transmission intensity. For this work, we used $k_E = 0.3$ when the baseline microfilarial prevalence of the modelled settings indicated meso- to hyperendemicity (≥40% but <80%), and $k_E = 0.4$ for holoendemicity (≥80%)[23]. We modelled the temporal dynamics of the microfilaricidal and embryostatic effects of ivermectin on *O. volvulus*[30] as well as a permanent sterilising effect of treatment on female adult worms[31]. To simulate MDA, therapeutic coverage levels (the proportion of individuals in the total population receiving treatment at each round) were specified, and the value of parameter $\rho$—measuring the correlation between rounds attended by eligible individuals (and reflecting the proportion of individuals who have never taken treatment after a given number of MDA rounds)—was estimated[32]. Supplementary Text 1 and Supplementary Table 1 provide further EPIONCHO-IBM details.

## Integration of OSD sequelae into EPIONCHO-IBM

We differentiate OSD sequelae into reversible (severe itch and RSD) and irreversible (ATR, DPM, HG) (see **Onchocerciasis skin disease (OSD)** above). APOD, CPOD and LOD are modelled as a single RSD condition[33]. Figure 1 depicts a schematic of the approach taken to model OSD in EPIONCHO-IBM, with daily probabilities of developing OSD sequelae given in Table 1 (Supplementary Text 2 provides further OSD modelling details).

## Comparison of modelled OSD age-prevalence profiles to baseline data

Modelled OSD age-prevalence profiles were compared with those calculated from the Kaduna data[26]. Using EPIONCHO-IBM[23], an annual biting rate (ABR, no. bites/person/year) of 615 was necessary to reproduce the approximately 50% baseline microfilarial prevalence in Kaduna. We generated microfilarial and OSD age-prevalence profiles for ABR = 240 and 20,000, respectively, for the lowest (23%, hypoendemic) and highest (85%, holoendemic) microfilarial prevalence (in those aged ≥5 years) in the study area[34]. For each microfilarial prevalence, we conducted 1000 simulations and calculated their mean and 95% UIs (2.5th–97.5th quantiles of stochastic model predictions). Age-prevalence estimates[26] were plotted with Clopper–Pearson, exact 95% confidence intervals (95% CIs)[35].

## Integration of OOD sequelae into EPIONCHO-IBM

Figure 2 provides a schematic of the approach used for modelling OOD. The probability of developing blindness is related log-linearly to microfilarial count 2 years in the past, following Eq. (2). Supplementary Text 3 provides further OOD modelling details.

## Comparison of modelled OOD age-prevalence profiles to baseline data

We compared our modelled OOD age-prevalence profiles with data from two sources. The first were data on age-stratified blindness and visual damage (impairment) prevalence for 12,100 individuals (aged ≥5 years) examined for visual acuity in 53 OCP villages prior to vector control[36]. Blindness was defined as the inability to count fingers at 1 m[36]. The ABR for these villages was estimated using data on microfilarial prevalence reported for 13,332 individuals aged ≥5 years who underwent skin-snip microscopy in 66 OCP villages (across Burkina Faso, Côte d'Ivoire, Ghana, Mali, Niger and Togo)[37], of which the 53 villages assessed for OOD were a subset. An ABR of 2000 bites/person/year (for an overall 68% microfilarial prevalence) was used. We calculated 95% UIs as described above, and generated microfilarial and OOD age-prevalence profiles for ABR values of 285 (30% microfilarial prevalence) and 60,000 bites/person/year (90%) to reflect the range of endemicities (from hypo- to holoendemic) recorded[37]. Supplementary Fig. 3 presents the microfilarial prevalence age profile (we kept the value of parameter $k_E = 0.3$ to facilitate comparison). The second source was OOD baseline data collected from 6831 individuals (aged ≥5 years) examined (for visual impairment and blindness according to WHO definitions)[38], in the same 34 Kaduna savannah villages of the OSD study[26]. We used the same ABR values described above in **Comparison of modelled OSD age-prevalence profiles to baseline data**. For each microfilarial prevalence, we conducted 1000 simulations and calculated their mean and 95% UIs. Age-prevalence estimates[36,38] were plotted with Clopper–Pearson 95% CIs[35]. We used the demographic structure data in the OSD study[26] to calculate the numbers examined for each age group in the OOD study[38].

## Impact of ivermectin MDA on OSD and OOD prevalence

**Modelling the trends of OSD prevalence under MDA.** We modelled the baseline epidemiological conditions and MDA programmatic features in seven study sites (5193 individuals aged ≥5 years) across Cameroon, Nigeria, Sudan and Uganda, for which the prevalence of severe itch, RSD and DPM were measured at baseline and 5–6 years into MDA[39]. The endemicity of the study sites at baseline was determined using Rapid Epidemiological Mapping of Onchocerciasis (REMO, a rapid epidemiological assessment method that uses the prevalence of onchocercal nodules—where adult *O. volvulus* worms reside—in samples of 30–50 adult men to identify at-risk communities[40]). Two cross-sectional surveys were conducted[39]. The first was done in 1998–1999, before the implementation of community-directed treatment with ivermectin (CDTI) in 1999–2000. We converted nodule prevalence (in those aged ≥20 years) into microfilarial prevalence (in those aged ≥5 years)[41,42] (Supplementary Text 4). Supplementary Fig. 4 shows the resulting

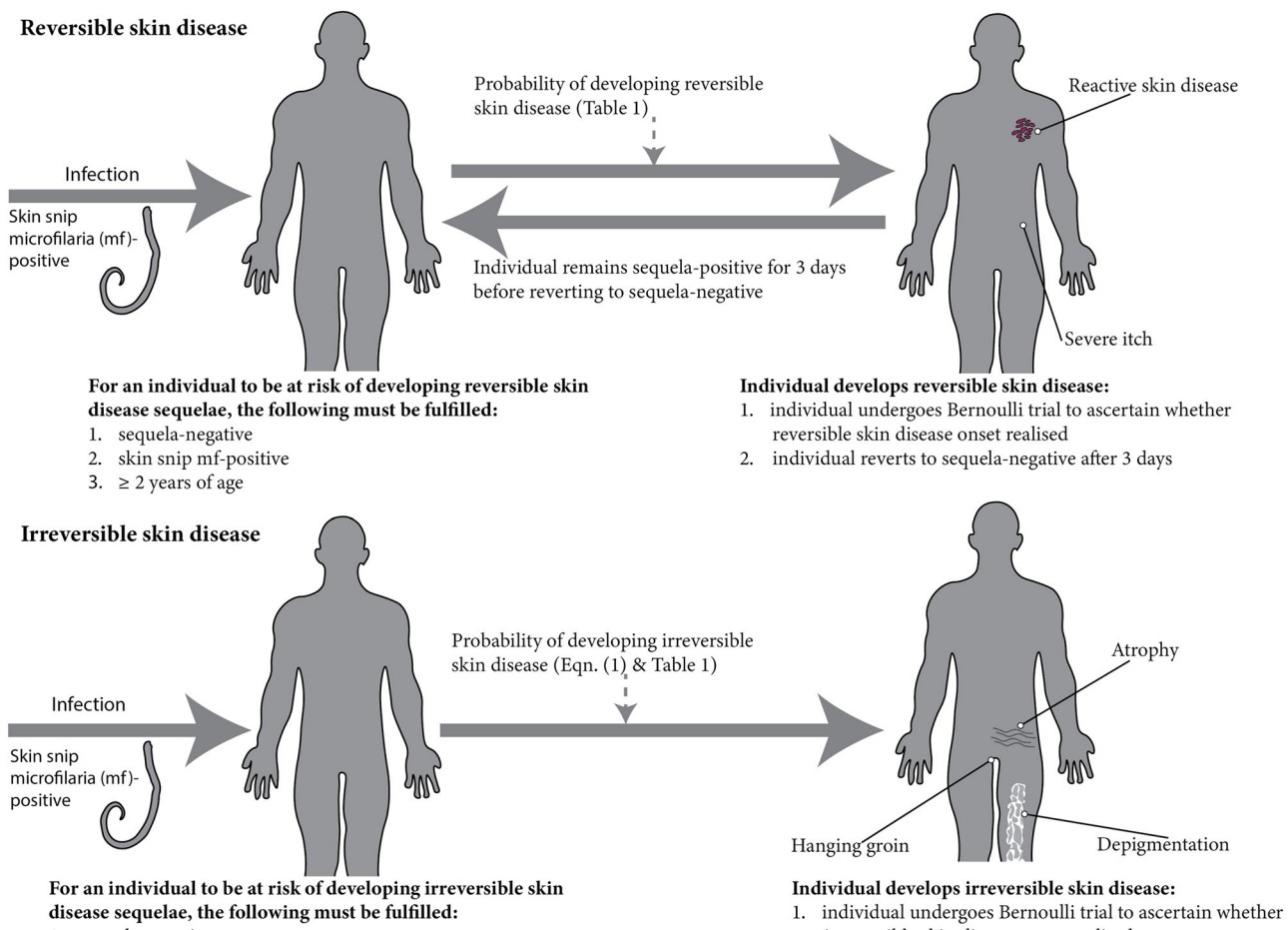

**Fig. 1 | Approach used to model onchocerciasis skin disease (OSD) sequelae in EPIONCHO-IBM.** OSD sequelae were categorised as reversible or irreversible. Reversible OSD includes severe itch and reactive skin disease (RSD, comprising acute papular onchodermatitis (APOD), chronic papular onchodermatitis (CPOD) and lichenified onchodermatitis (LOD), as a single condition). We assumed that after developing the condition, individuals remain positive for severe itch or RSD for 3 days, after which they revert and become eligible again to develop the condition; durations ranging 1–5 days were tested, with 3 days being the most consistent with the severe itch and RSD age-prevalence profiles shown in Fig. 3. Irreversible OSD (Eq. (1)) includes skin atrophy (ATR), depigmentation (DPM, either mild or severe), and hanging groin (HG). The probabilities of developing OSD were derived using data from Murdoch et al.[26] (Table 1).

(posterior) microfilarial prevalence distributions for each site. Baseline (median) microfilarial prevalence values were used to estimate ABRs for modelling pre-control conditions. The second cross-sectional survey (5180 individuals aged ≥5 years) was conducted 5–6 years into the CDTI programme in 2004–2005[39].

The CDTI programme for each site was modelled using the coverage of total population values reported for each round and the never-treated proportion after 5–6 rounds[39], which allowed estimation of parameter $\rho$[32]. In Taraba (Nigeria) and Bushenyi (Uganda), the never-treated proportion remained high (25% in Taraba and 34% in Bushenyi, after 5 rounds). To mirror this, a proportion of eligible individuals were randomly assigned to never receiving treatment from round 1. The values of this proportion were calculated as 6% in Taraba and 17% in Bushenyi. Parameter $\rho$ was calculated for the remaining eligible population (Table 2). Supplementary Fig. 5 shows the modelled never-treated proportion compared to data for the seven study sites. The mean coverage values across the first 5–6 years of treatment[39] were used to continue simulating MDA until 2020. Table 2 summarises the baseline epidemiological conditions and parameter inputs to simulate CDTI programmatic features in the seven sites. For each of these, we conducted 1000 simulations and calculated their mean and 95% UIs. Converted microfilarial (median) prevalence for each site and survey was plotted with 95% credible intervals (95% CrIs), based on the posterior distributions of

microfilarial prevalence converted from nodule prevalence[41,42]. The trajectories of the recorded OSD sequelae (severe itch, RSD, DPM) were modelled for each site and plotted together with prevalence estimates[39] and Clopper–Pearson 95% CIs[35].

**Modelling the trends of OOD under MDA.** To assess the impact of MDA on OOD, we used two cross-sectional studies measuring blindness prevalence in savannah settings before and after a period of ivermectin MDA. The first was conducted in the village of Gami, Central African Republic (CAR), where 309 (of a total of 430) individuals aged ≥5 years were examined for microfilaridermia and 301 for visual acuity in 1990 (before MDA). Five years later (after five rounds), 362 (of 451) individuals aged ≥5 years underwent skin-snip microscopy, and 346 were ophthalmologically examined in 1995[43]. Individuals who had a visual acuity of <3/10 (or 6/18) but could see enough not to need help in their normal daily activities were considered 'visually impaired'. Those who could see nothing or not enough to cope alone with their normal daily activities were considered 'functionally blind'[43]. The second study took place in the village of Galadimawa, Kaduna State (one of the 34 villages evaluated in northern Nigeria)[26,27,38]. The prevalence of microfilaridermia was assessed in 1987[34] in 671 individuals aged ≥5 years. The prevalence of blindness (by examination of the visually disabled, EVD) was evaluated in

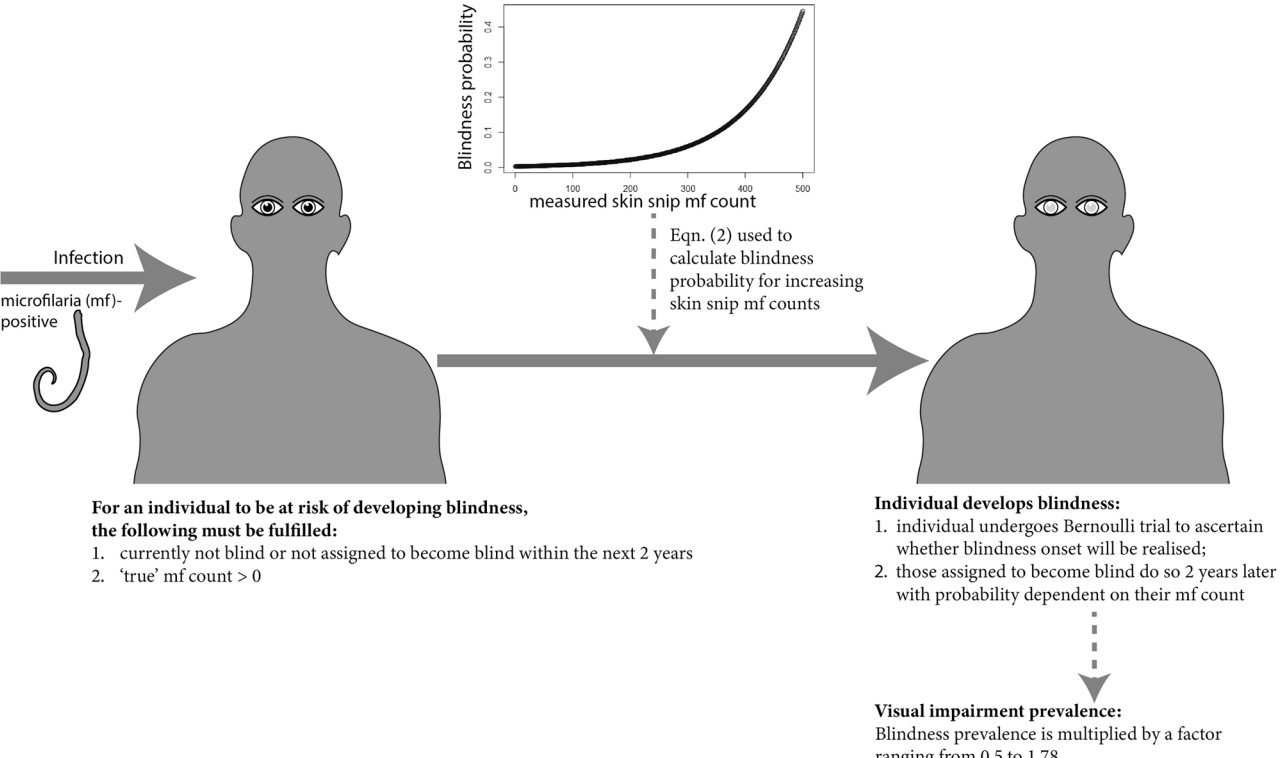

**Fig. 2 | Approach used to model onchocerciasis ocular disease (OOD) in EPIONCHO-IBM.** Blindness was modelled as an irreversible sequela. Individuals are at risk of developing blindness based on their modelled 'true' rather than 'detectable' microfilarial status. This enables the model to generate small probabilities in those individuals with false-negative skin snips. The probability of blindness (Eq. (2)) is log-linearly related to detectable microfilarial count lagged by 2 years (i.e., microfilarial count 2 years in the past is related to current probability of developing blindness), adapted from Little et al.[28], who found no evidence of statistically significant differences between males and females in blindness incidence. Visual impairment prevalence was calculated by multiplying blindness prevalence by 0.5, 1.0 and 1.78[22,29].

1988 (in 711 individuals out of the total population of 716), before the start of CDTI in 1991[34] and again in 2016 (in an estimated 1419 individuals), after 25 treatment rounds[44]. As Galadimawa was one of the villages selected to participate in the trial investigating the effect of ivermectin treatment on optic nerve disease[27], some individuals received treatment since 1989 as part of the trial[27,44].

The study in Gami provided information on baseline microfilarial prevalence, treatment coverage and proportion of eligible individuals who refused treatment (never treated)[43], necessary to parameterise EPIONCHO-IBM. For Galadimawa[44], additional data sources were necessary, including microfilarial prevalence and coverage of the total population (in Kauru Local Government Area, where Galadimawa is located)[34]. To estimate the never-treated proportion of the eligible population, two ivermectin compliance studies conducted in Nigeria were used[45,46]. Table 3 provides baseline epidemiological conditions and parameter inputs to simulate programmatic features of ivermectin MDA in the study sites of CAR[43] and Nigeria[44] used to model blindness trends under MDA. The mean coverage values across five years of treatment in CAR[43], or 17 years in Nigeria[34] were used to continue simulating MDA until 2020. For each site, we conducted 1000 simulations and calculated their mean and 95% UIs. The trends in blindness prevalence were modelled and plotted together with estimates from the data[43,44] and Clopper-Pearson 95% CIs[35].

Data analysis and plotting were conducted using R statistical software (v4.3.2)[47]. Transmission dynamics modelling was conducted using Python programming software (v3.10)[48]. We adhered to the five principles of the Neglected Tropical Diseases (NTD) Modelling Consortium regarding Policy-Relevant Items for Reporting Models in Epidemiology of NTDs (PRIME-NTD), for good practice in NTD modelling[49] (Supplementary Text 5, Supplementary Table 2).

## Ethical approval
This study did not require approval from an institutional review board (IRB) because it was based exclusively on publicly available data and did not involve any identifiable human subjects or personal information.

## Results
### Modelled and observed OSD age-prevalence profiles
Figure 3 presents modelled and observed[26] microfilarial and OSD age-prevalence profiles at baseline for the range of endemicities simulated (23, 50 and 85% overall microfilarial prevalence). The modelled reversible OSD sequelae (severe itch and RSD) increase rapidly with age, with a decreased rate after age 20 years, mirroring microfilarial prevalence. The modelled irreversible OSD sequelae (ATR, DPM, HG) increase gradually and monotonically with age. The modelled mean prevalence and 95% UIs capture the data within the range of endemicities but underestimate the prevalence of ATR and HG for the oldest age group.

### Modelled and observed OOD age-prevalence profiles
Figure 4 compares the modelled pre-intervention OOD age-prevalence profiles with those from baseline OCP[36] and Kaduna data[38]. Both model- and data-derived OOD age-prevalence profiles increase monotonically with age (as for irreversible OSD). Modelled OOD age-prevalence profiles are presented for overall microfilarial prevalences of 30, 68 and 90% for the OCP data[37] and of 23, 50 and 85% for the Kaduna data[26,34].

### Modelled and observed trends of microfilarial and OSD prevalence under MDA
Figure 5 presents the trends of microfilarial prevalence under CDTI programmatic conditions in each of the seven study sites, summarised in

**Table 2 | Baseline epidemiological conditions and parameter inputs to simulate programmatic features of ivermectin mass drug administration (MDA) in seven study sites[39] for modelling onchocerciasis skin disease (OSD) trajectories under MDA**

| Country Site | Sample size (N)[a] | Observed pre-control nodule prevalence[a] (95% CI) (%) | Converted (median) microfilarial prevalence[b] (95% CrI) (%) | Simulated prevalence[c] (%) (Modelled ABR[b], bites/person/yr) | Coverage of total population for each MDA round[a] (%) | Proportion of never-treated population after n MDA rounds[a] (%) | Parameter $\rho$[d] (assigned proportion of never-treated population[e]) (%) | Mean[a] total population coverage (for remaining years) (%) |
|---|---|---|---|---|---|---|---|---|
| **Cameroon** | | aged ≥20 yr | aged ≥5 yr | | n = 6 (1999-2004) | | | |
| Kumba | 764 | 44.4 (40.8-48.0) | 62.9 (32.3-85.8) | 63.1 (1300) | 76, 56, 23, 23, 48, 65 | 15 | 0.355 (0) | 48 |
| Ngambe | 739 | 41.5 (38.0-45.2) | 60.9 (30.4-84.8) | 60.6 (1110) | 37, 39, 26, 52, 62, 66 | 13 | 0.24 (0) | 48 |
| **Nigeria** | | aged ≥20 yr | aged ≥5 yr | | n = 5 (1999-2003 or 2000-2004) | | | |
| Cross River | 759 | 28.1 (24.9-31.4) | 50.3 (22.2-78.5) | 50.3 (615) | 73, 73, 76, 78, 77 | 19 | 0.8 (0) | 75 |
| Kogi | 755 | 40.3 (36.7-43.9) | 60.2 (29.2-84.2) | 60.4 (1,082) | 77, 77, 79, 80, 79 | 19 | 0.9 (0) | 78 |
| Taraba | 662 | 13.3 (10.8-16.1) | 33.8 (12.5-64.0) | 33.5 (320) | 80, 82, 84, 85, 81 | 25 | 0.999 (6) | 82 |
| **Sudan** | | aged ≥20 yr | aged ≥5 yr | | n = 5 (2000-2004) | | | |
| Raja | 756 | 63.1 (59.5-66.5) | 73.2 (43.0-90.6) | 73.6 (3250) | 77, 64, 60, 42, 67 | 6 | 0.1 (0) | 62 |
| **Uganda** | | aged ≥20 yr | aged ≥5 yr | | n = 5 (1999-2003) | | | |
| Bushenyi | 758 | 13.2 (10.8-15.8) | 33.7 (12.4-65.0) | 33.5 (320) | 68, 79, 78, 75, 78 | 34 | 0.9 (17) | 75 |

[a]Values reported in Ozoh et al.[39].
[b]Following the method in Coffeng et al.[41], and Coffeng[42].
[c]Using $k_E$ = 0.3 in EPIONCHO-IBM[23].
[d]$\rho$ measures the strength of individual-level correlation between attending successive treatment rounds, ranging between 0 (individuals are randomly assigned to receiving or not receiving treatment at each round) and 1 (individuals assigned to receiving or not receiving treatment in round 1 will always receive (or not receive) treatment in successive rounds, indicating fully systematic non-adherence)[32].
[e]Fixed never-treated proportion randomly assigned from round 1 to eligible population that was necessary to attain the proportion of those never-treated recorded after 5 or 6 rounds.

**Table 3 | Baseline epidemiological conditions and parameter inputs to simulate programmatic features of ivermectin mass drug administration (MDA) in two study sites[34,43] for modelling onchocerciasis ocular disease (OOD) trajectories under MDA**

| Country Site | Sample size (N)[a] | Observed microfilarial prevalence[a] (95% CI) (%) | Simulated prevalence[b] (Modelled ABR[b], bites/person/yr) | Coverage of total population for each MDA round[a] (%) | Proportion of never-treated population after n MDA rounds[c] (%) | Parameter $\rho$[d] (additional proportion of never-treated population[e]) (%) | Mean[a] total population coverage (for remaining years) (%) |
|---|---|---|---|---|---|---|---|
| **Central African Republic[43]** | | aged ≥5 yr | | n = 5 (1990-1994) | | | |
| Gami | 309 | 88.0 (83.2-90.9) | 88.3 (14,000) | 72, 75, 75, 75, 77 | 1[43] | 0 (0) | 75 |
| **Nigeria[34]** | | aged ≥5 yr | | n = 8[45]-17[46] (1991-1998; 1991-2007) | | | |
| Galadimawa | 671 | 46.8 (43.0-50.7) | 47.3 (530) | 76.2 (74.4-80.3) | 6[45]-11[46] | 0.3[45]-0.65[46] (0) | 76 |

[a]Values reported in Kennedy et al.[43], and Tekle et al.[34].
[b]Using $k_E$ = 0.4 for CAR and $k_E$ = 0.3 for Nigeria in EPIONCHO-IBM[23].
[c]Values from Brieger et al.[45] and Osue et al.[46].
[d]$\rho$ measures the strength of eligible individual-level correlation between attending successive treatment rounds, ranging between 0 (individuals are randomly assigned to receiving treatment at each round) and 1 (individuals assigned to receiving (or not) receiving treatment in round 1 will always receive (or not receive) treatment in successive rounds, indicating fully systematic non-adherence)[32].
[e]Additional never-treated proportion of eligible population required to attain the proportion of those never-treated after the specified number of MDA rounds.

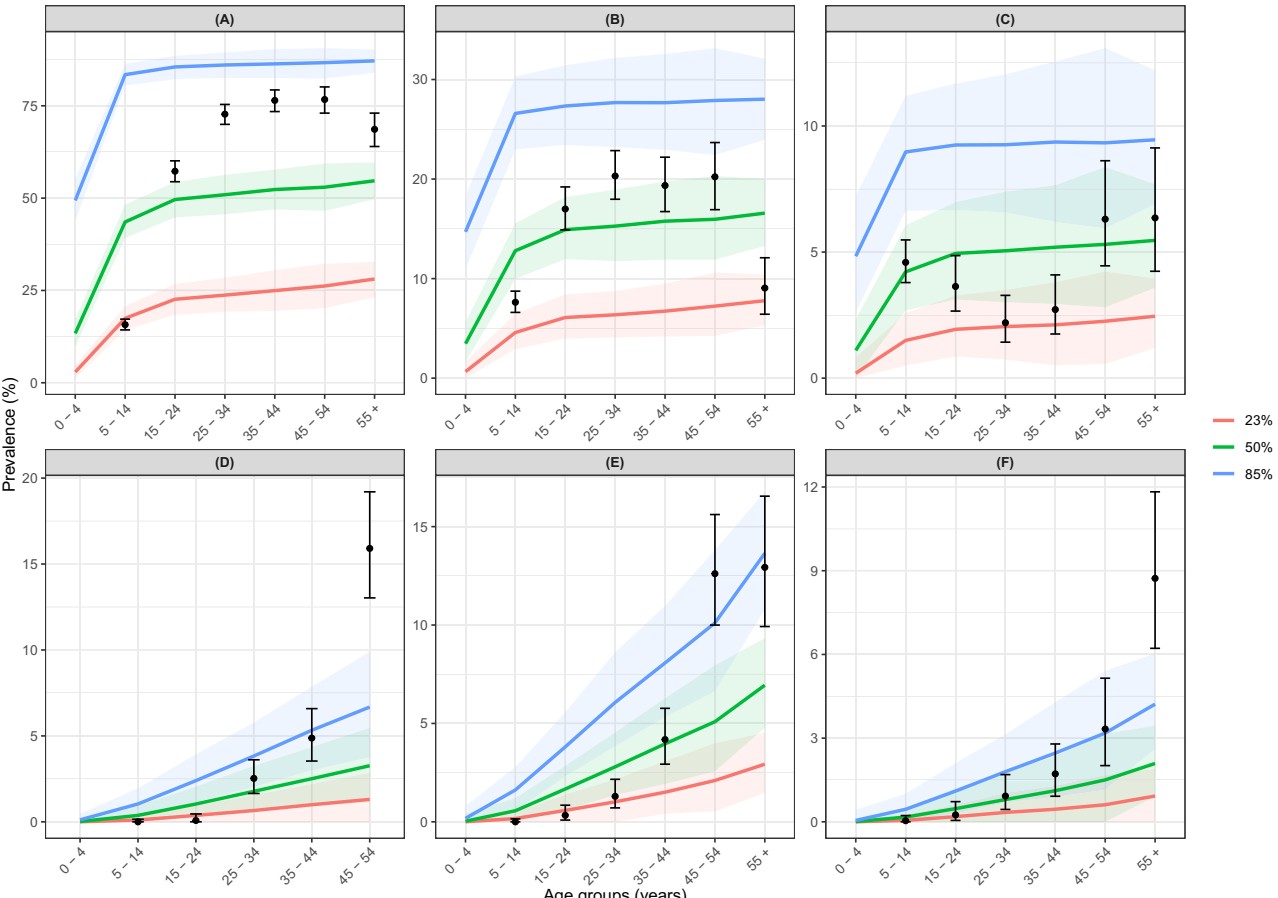

**Fig. 3 | Modelled and observed *Onchocerca volvulus* microfilarial and oncho-cerciasis skin disease (OSD) age-prevalence profiles.** Age-prevalence profiles were generated using EPIONCHO-IBM with annual biting rate (ABR) of 240, 615 and 20,000 bites/person/year and individual exposure parameter $k_E$ = 0.3 for, respectively, baseline microfilarial prevalence of 23, 50 and 85% for Kaduna, Nigeria (Murdoch et al.[26], *n* = 6643; Tekle et al.[34], *n* = 3703). **A** Skin microfilariae. **B** Severe itch. **C** RSD reactive skin disease. **D** ATR skin atrophy. **E** DPM depigmentation.

**F** HG hanging groin. In (**D**), data and model outputs are not presented for individuals aged ≥55 years; ATR was not scored in individuals aged ≥50 years[26] to avoid confusion with senile skin atrophy. Solid lines are the mean of 1000 model runs; shaded areas are the 95% uncertainty intervals (2.5th–97.5th quantiles of stochastic simulations); black circles are prevalence estimates from the data[26] with Clopper-Pearson 95% confidence intervals[35]. NB: *y*-axis in different scales.

Table 2[39]. (Converted) microfilarial prevalence after 5 or 6 years of CDTI was well captured by the model in Kumba (Cameroon), Cross River (Nigeria), Raja (Sudan) and Bushenyi (Uganda). In Ngambe (Cameroon), and Kogi and Taraba (Nigeria), the model captured the data less well, albeit still within the 95% UIs; infection prevalence 5–6 years into the programme had hardly changed from that recorded at baseline for these sites, in contrast to the decreases predicted by EPIONCHO-IBM.

Figure 6 shows the modelled and observed prevalence trajectories for severe itch. The modelled trajectories captured the general trends observed in Kumba, Cross River and Raja. In Ngambe and Kogi, the observed prevalence of severe itch increased, rather than decreased (as predicted by the model) during the first 5-6 years of CDTI. The model overestimated the prevalence of severe itch in Taraba and underestimated it in Bushenyi.

Figure 7 presents the modelled and observed trajectories of RSD prevalence. EPIONCHO-IBM captured the magnitude and decline in Taraba. In Cross River, the model underestimated the baseline prevalence of RSD but reproduced the value observed after 5 years of CDTI. Conversely, in Kogi, the model captured well the baseline RSD prevalence, but in this site, RSD increased rather than decreased, whilst EPIONCHO-IBM predicted that it would decline by 60%. In the remaining sites, there were dramatic relative decreases (52–91%) in the prevalence of RSD (with the exception of Bushenyi, where the decrease was less pronounced (20%) albeit statistically significant as indicated by the 95% CIs), but the model generally

underestimated both the baseline RSD prevalence and the relative reductions under CDTI (28–64%).

Figure 8 shows the modelled and observed trends for DPM prevalence (the only irreversible dermatological sequela followed up[39]). EPIONCHO-IBM captured well the observed trajectories in Bushenyi, with the prevalence for Cross River, Kogi, and Raja mostly falling within the model 95% UIs. In Kumba and Ngambe, the model underestimated observed trends, and slightly overestimated them in Taraba. In Ngambe, Taraba and Raja, the observed prevalence had increased, albeit non-significantly.

**Modelled and observed trends of microfilarial and OOD prevalence under MDA**

Figure 9 presents the modelled and observed trends of microfilarial and blindness prevalence in the holoendemic village of Gami (CAR)[43] and the mesoendemic village of Galadimawa (Kaduna State, Nigeria)[44]. The reported reduction in microfilarial prevalence is not captured in CAR after 5 CDTI rounds (Fig. 9A), but it is well captured after 17 rounds in Nigeria[34] (Fig. 9B). Modelled blindness prevalence declined in both CAR and Nigeria, although not as substantially as indicated by the data after 5 treatment rounds in CAR and 25 rounds in Nigeria. Despite using $k_E$ = 0.4 for Gami (which led to an ABR of 14,000 bites/person/year and less severe density dependence[23]), EPIONCHO-IBM overestimated microfilarial and blindness prevalence 5 years into MDA (Fig. 9A, C). However, the upper 95%

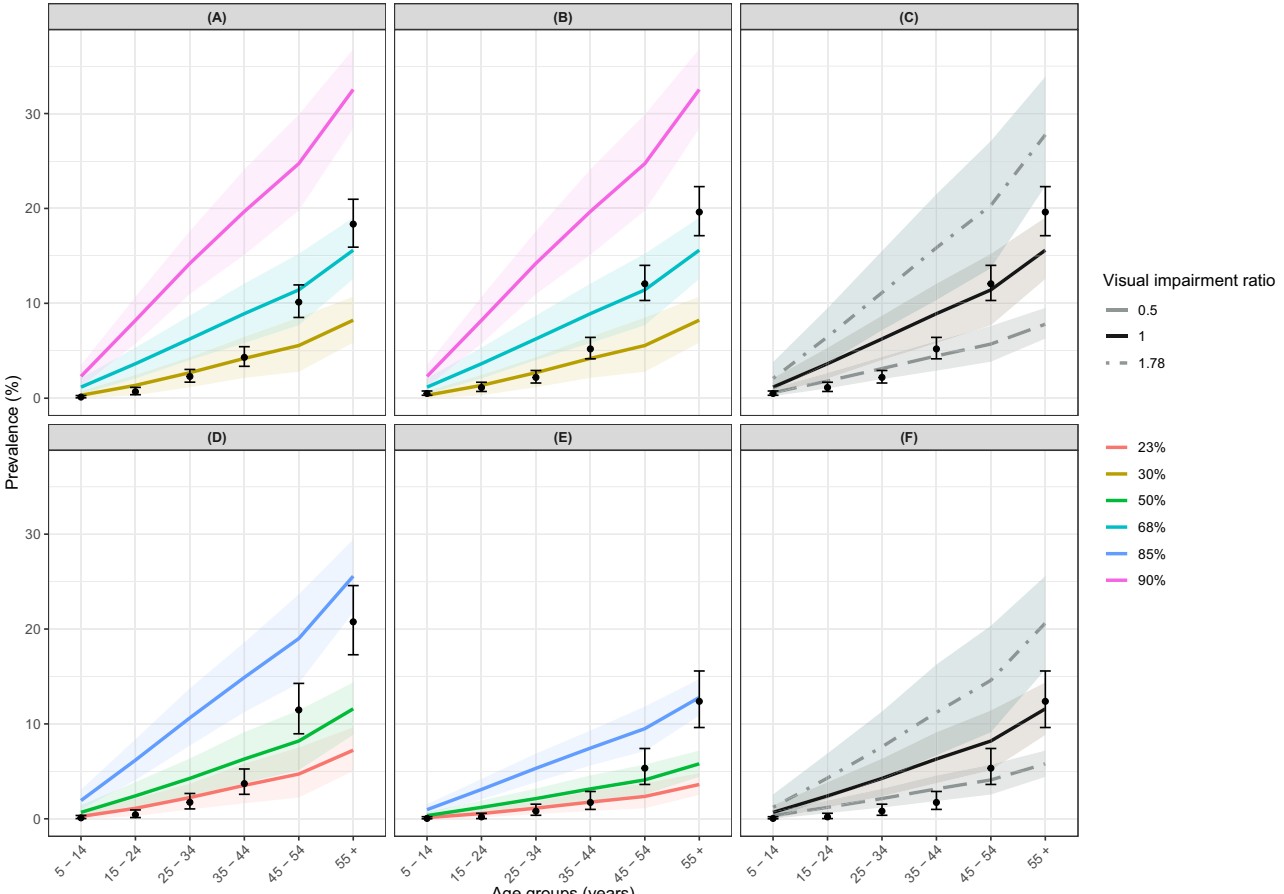

**Fig. 4 | Modelled and observed onchocerciasis ocular disease (OOD) age-prevalence profiles.** Age-prevalence profiles were generated using EPIONCHO-IBM with annual biting rate (ABR) of 285, 2000 and 60,000 bites/person/year and individual exposure parameter $k_E = 0.3$ for, respectively, microfilarial prevalence of 30, 68 and 90% for the Onchocerciasis Control Programme in West Africa data (Kirkwood et al.[36,37], $n = 12,100$, upper panels) and 240, 615 and 20,000 bites/person/year and individual exposure parameter $k_E = 0.3$ for, respectively, microfilarial prevalence of 23, 50 and 85% for the Kaduna data (Abiose et al.[38], $n = 6827$; Tekle et al.[34], $n = 3703$, lower panels). **A**, **D** Blindness. **B**, **E** Visual impairment. **C**, **F** Visual impairment for ratios of visual impairment to blindness of 0.5, 1.0 and 1.78. Solid lines are the mean of 1000 model runs; shaded areas are the 95% uncertainty intervals (2.5th to 97.5th quantiles of stochastic simulations); black circles are the prevalence estimates from the data[38] with Clopper-Pearson 95% confidence intervals[35].

limit of the latter fell into our 95% UI. For Galadimawa, EPIONCHO-IBM underestimated the reported reduction in blindness prevalence, Fig. 9D).

## Discussion

We have integrated skin and ocular morbidities into the onchocerciasis transmission model, EPIONCHO-IBM, expanding our previous work[22,25]. We used extensive baseline data, collected prior to the implementation of interventions, for calibrating the model to pre-control transmission conditions. The parameterisation of the probabilities of developing OSD and OOD sequelae was also derived from pre-intervention datasets. Rather than outdated, such data remain highly relevant, robust, and essential given the lack of recent, large-scale pre-control morbidity surveys.

The observed age-profiles of microfilarial prevalence are encompassed within the endemicity range modelled (ABR between 240 and 20,000 bites/person/year). For the highest ABR (overall microfilarial prevalence of 85%), the roughly 50% microfilaridermia prevalence in the under 5-year-olds is comparable with values reported in holoendemic savannah areas of northern Cameroon (where the prevalence in the 3–9-year-olds exceeded 60%[50]) and CAR (where the prevalence among 5–9-year-olds was 67%[43]). This high predicted prevalence in young children results from the EPIONCHO-IBM sex- and age-dependent exposure functions, according to which children are exposed from birth[51]. Hence, a 2–3% blindness prevalence is predicted in the 5–14-year-olds for high transmission intensities.

In a holoendemic savannah area of northern Cameroon with 93% (85–99%) microfilarial prevalence, punctate keratitis was observed in 10% of children (both sexes) and sclerosing keratitis in 2% of boys aged 5–9 years[52].

Our modelling approach for skin disease linked the probability of developing OSD sequelae to infection status (being mf-positive) rather than to infection intensity (microfilarial load), because in the dataset we used[26], the latter was not particularly informative. This approach—for both reversible (severe itch; RSD) and irreversible (ATR; DPM; HG) OSD—relies on the broad assumption that prevalence is proportional to incidence (for reversible conditions) or to cumulative incidence (for irreversible conditions). In both cases, this simplification assumes stable transmission dynamics, age-independent risk, and negligible competing risks (e.g., from death or migration)—conditions that may not fully hold, but which nonetheless yield reasonable approximations under endemic equilibrium and limited data. A more comprehensive alternative would involve a dynamic model of morbidity risk, incorporating estimates of age-specific force of infection derived from age-stratified infection prevalence data[53]. This would avoid the need to assume constant exposure over age, constant morbidity risk, or linear accumulation of morbidity, and would instead allow infection and morbidity to be modelled explicitly as age-structured processes under more flexible assumptions.

Another potential refinement could focus on modelling APOD, CPOD and LOD separately. We tested durations of 1–5 days for mf-positive

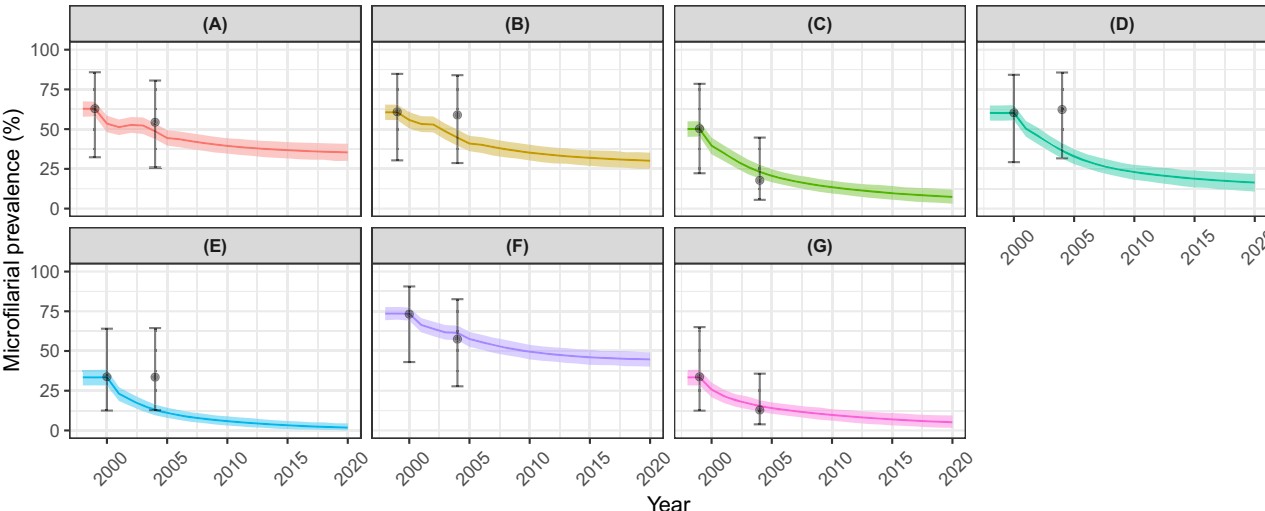

**Fig. 5 | Modelled and observed impact of mass drug administration (MDA) delivered as community-directed treatment with ivermectin (CDTI) on *Onchocerca volvulus* microfilarial prevalence across seven study sites assessed for onchocerciasis skin disease.** Cameroon: **A** Kumba (1st timepoint *n* = 764; 2nd timepoint *n* = 858). **B** Ngambe (*n* = 739; *n* = 752). Nigeria: **C** Cross River (*n* = 759; *n* = 647). **D** Kogi (*n* = 755; *n* = 783). **E** Taraba (*n* = 662; *n* = 615). Sudan: **F** Raja (*n* = 756; *n* = 770). Uganda: **G** Bushenyi (*n* = 758; *n* = 755). The data are from Ozoh et al.[39], with modelled annual biting rates and programmatic parameters as in Table 2, and individual exposure parameter $k_E$ = 0.3. Solid lines are the mean of 1000 model runs; shaded areas are the 95% uncertainty intervals (2.5th–97.5th quantiles of stochastic simulations); black circles are the (median) microfilarial prevalence values converted from nodule prevalence data[39,41,42] with 95% credible intervals.

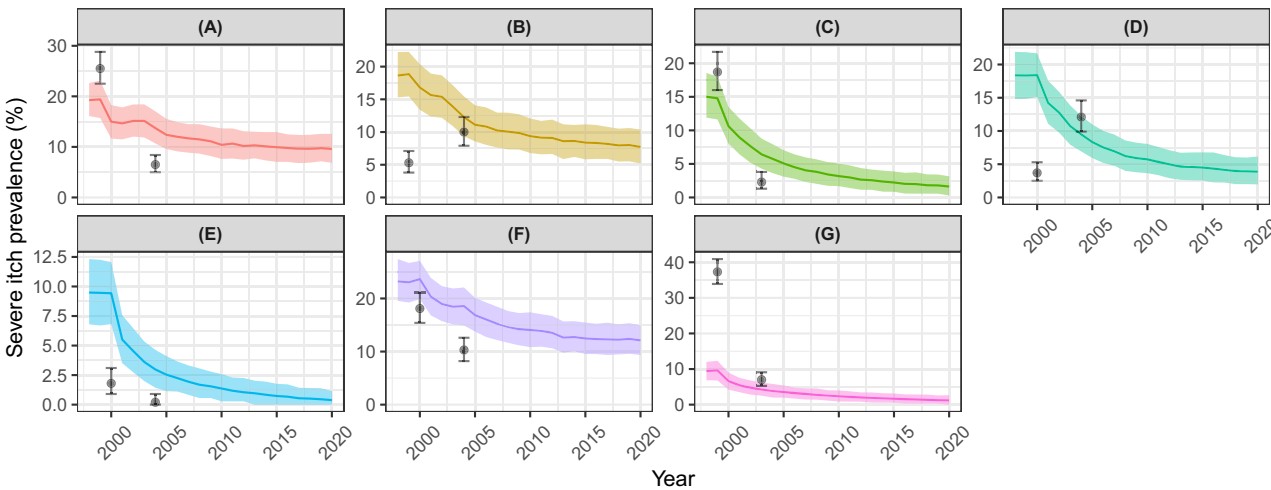

**Fig. 6 | Modelled and observed impact of mass drug administration (MDA) delivered as community-directed treatment with ivermectin (CDTI) on severe itch prevalence across seven study sites assessed for onchocerciasis skin disease.** Cameroon: **A** Kumba (1st timepoint, *n* = 764; 2nd timepoint, *n* = 858). **B** Ngambe (*n* = 739; *n* = 752). Nigeria: **C** Cross River (*n* = 759; *n* = 647). **D** Kogi (*n* = 755; *n* = 783). **E** Taraba (*n* = 662; *n* = 615). Sudan: **F** Raja (*n* = 756; *n* = 770). Uganda: **G** Bushenyi (*n* = 758; *n* = 755). The data are from Ozoh et al.[39], with modelled annual biting rates and programmatic parameters as in Table 2, and individual exposure parameter $k_E$ = 0.3. Solid lines are the mean of 1000 model runs; shaded areas are the 95% uncertainty intervals (2.5th–97.5th quantiles of stochastic simulations); black circles are the severe itch prevalence estimates from the data[39] with Clopper–Pearson 95% confidence intervals[35]. NB: *y*-axis in different scales.

individuals to remain in the severe itch and RSD conditions before reverting and regaining their susceptibility to developing such conditions, and selected 3 days as the most consistent to reproduce the (combined) observed age-prevalence profiles. However, although APOD might last for several days, CPOD and LOD likely last for longer. LOD is characterised by pruritic, hyperpigmented, hyperkeratotic plaques, usually in an asymmetrical distribution involving one limb (also known as 'sowda')[13]. LOD results from heightened host immunological responses, which kill microfilariae at the expense of substantial skin pathology[17]. It is conceivable that the OSD sequelae within RSD reflect a spectrum of variation in immunological responses rather than being part of a progressive, multistage condition. Also, the probability of developing reversible OSD may not be the same between

individuals who have previously developed it and those who develop it for the first time. Currently, the natural histories of individual forms of RSD are poorly understood, and suitable longitudinal cohort studies to adequately quantify their temporal dynamics are lacking.

We did not explicitly model the probability of becoming visually impaired, but examined the ratio of visual impairment to blindness in the OCP[36] and Kaduna[38]. In the former, this ratio was approximately equal to 1, while in the latter it was about 0.5 (partly owing to differences in the criteria used to define visual impairment). We used these ratios to model visual impairment in each setting. Our modelled OOD age-prevalence profiles are consistent with the data for all age groups within the endemicity ranges simulated. In our previous deterministic approach[22], we used a greater ratio

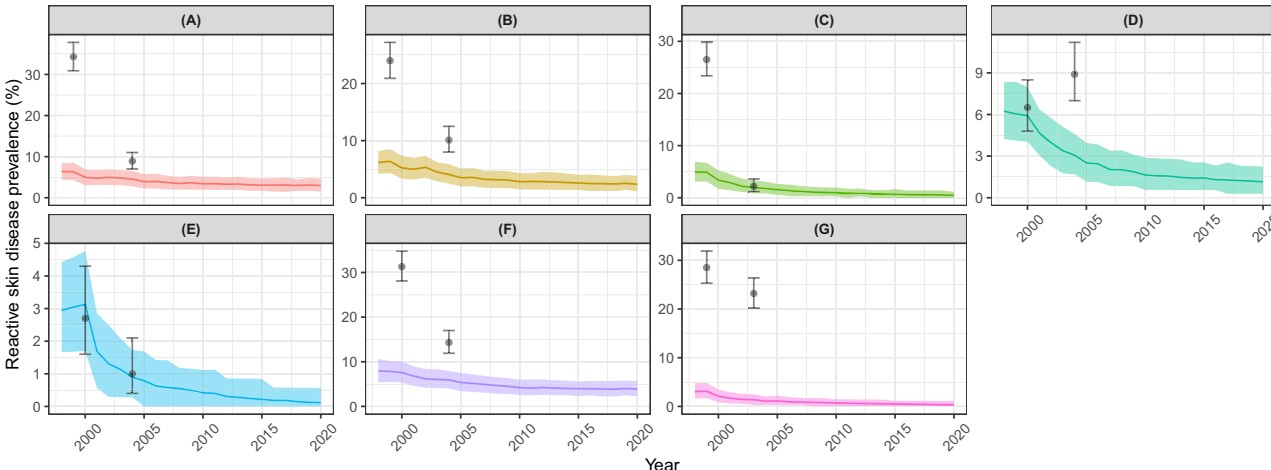

**Fig. 7 | Modelled and observed impact of mass drug administration (MDA) delivered as community-directed treatment with ivermectin (CDTI) on reactive skin disease (RSD) prevalence across seven study sites assessed for onchocerciasis skin disease.** Cameroon: **A** Kumba (1st timepoint, $n = 764$; 2nd timepoint, $n = 858$). **B** Ngambe ($n = 739$; $n = 752$). Nigeria: **C** Cross River ($n = 759$; $n = 647$). **D** Kogi ($n = 755$; $n = 783$). **E** Taraba ($n = 662$; $n = 615$). Sudan: **F** Raja ($n = 756$; $n = 770$).

Uganda: **G** Bushenyi ($n = 758$; $n = 755$). The data are from Ozoh et al.[39], with modelled annual biting rates and programmatic parameters as in Table 2, and individual exposure parameter $k_E = 0.3$. Solid lines are the mean of 1000 model runs; shaded areas are the 95% uncertainty intervals (2.5th–97.5th quantiles of stochastic simulations); black circles are the RSD prevalence estimates from the data[39] with Clopper-Pearson 95% confidence intervals[35]. NB: *y*-axis in different scales.

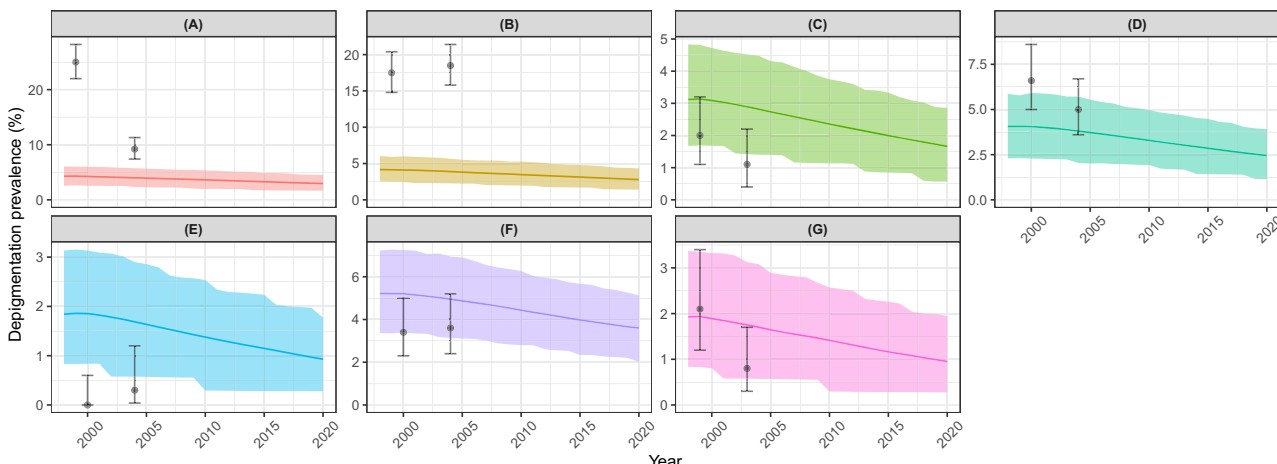

**Fig. 8 | Modelled and observed impact of mass drug administration (MDA) delivered as community-directed treatment with ivermectin (CDTI) on depigmentation (DPM) prevalence across seven study sites assessed for onchocerciasis skin disease.** Cameroon: **A** Kumba (1st timepoint, $n = 764$; 2nd timepoint, $n = 858$). **B** Ngambe ($n = 739$; $n = 752$). Nigeria: **C** Cross River ($n = 759$; $n = 647$). **D** Kogi ($n = 755$; $n = 783$). **E** Taraba ($n = 662$; $n = 615$). Sudan: **F** Raja ($n = 756$; $n = 770$).

Uganda: **G** Bushenyi ($n = 758$; $n = 755$). The data are from Ozoh et al.[39], with modelled annual biting rates and programmatic parameters as in Table 2, and individual exposure parameter $k_E = 0.3$. Solid lines are the mean of 1000 model runs; shaded areas are the 95% uncertainty intervals (2.5th–97.5th quantiles of stochastic simulations); black circles are the DPM prevalence estimates from the data[39] with Clopper–Pearson 95% confidence intervals[35]. NB: *y*-axis in different scales.

of visual impairment to blindness, of 1.78[29], but considerable uncertainty remains surrounding this ratio. In ONCHOSIM, the ratio of visual impairment to blindness in savannah settings was 0.8 (high hyperendemicity), 1.5 (hyperendemicity), and 3 (mesoendemicity), for an average of 1.77[33], in agreement with the value of 1.78[29].

The study in CAR did not provide age stratified visual impairment data[43], but a comparison with blindness prevalence (albeit with non-identical age groups) indicates that among 15–29-year-olds, 7.8% (95% CI = 3.2–15.4%) were blind, increasing to 24.2% (14.2–36.7%) among 30–45 year olds and 23.3% (9.9–42.3%) among ≥46-year-olds, prior to ivermectin MDA. These values are consistent with our projected blindness age-prevalence profiles for holoendemicity; Gami had a baseline microfilarial prevalence of 88%[43]. Our parameterisation of the probability of developing blindness was based on data from savannah settings[28]. Given that a recent re-appraisal of blindness prevalence in savannah and forest settings reported

no appreciable differences due to 'savannah' and 'forest' parasite 'strains'[54], we consider that our results are generalisable. We did not differentiate between the sexes when modelling the age-prevalence blindness profiles in the OCP[36] and Kaduna[38]. However, the overall blindness prevalence in males was twice that in females in both studies (4.8% vs. 2.4%, and 4.5% vs. 2.2%, respectively). In CAR, the prevalence was 1.6 times as high in males compared to females (11.5% vs. 7.4%)[43]. Future work will investigate the role of age- and sex-specific exposure[51] in these differences as previously done for OAE[25].

In contrast to OSD, we linked the probability of developing blindness to (2-year lagged) microfilarial load. As the natural life-expectancy of microfilariae is 1–2 years[50], with a maximum longevity of 2.5 years[23], this lag could represent a proxy for dying/dead microfilariae. In fact, clinical manifestations of onchocerciasis are deemed to be the result of cumulative host immunological reactions around dying microfilariae (against their

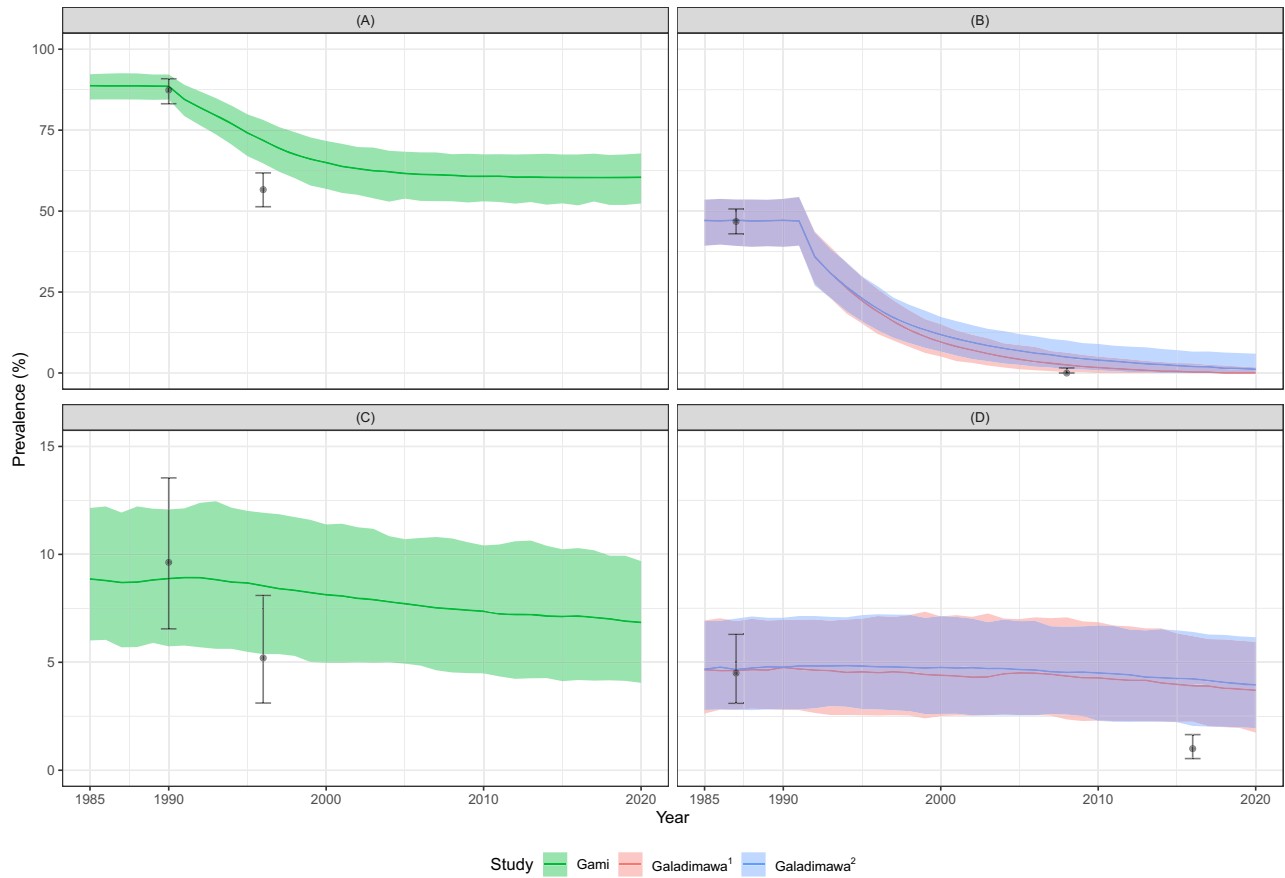

**Fig. 9 | Modelled and observed impact of mass drug administration (MDA) delivered as community-directed treatment with ivermectin (CDTI) on microfilarial and blindness prevalence in two study sites assessed for onchocerciasis ocular disease.** Central African Republic (Gami): **A**, **C**. Nigeria (Galadimawa): **B**, **D**. The prevalence of microfilaridermia (**A**, **B**) is presented at baseline (Gami, n = 309 (**A**); Galadimawa, n = 671 (**B**)) and at the time of parasitological evaluation (5 years into the programme in Gami[43], n = 362 (**A**); 17 years in Galadimawa[34], n = 235 (**B**)). The prevalence of blindness (**C**, **D**) is presented at baseline (Gami, n = 301 (**C**); Galadimawa, n = 711 (**D**)), and at the time of ophthalmological evaluation (5 years into the programme in Gami[43], n = 346 (**C**); 25 years in Galadimawa[44], n = 1419 (**D**)). Modelled annual biting rates and programmatic parameters are as in Table 3; individual exposure parameter $k_E = 0.4$ for Gami (a holoendemic setting) and 0.3 for Galadimawa (a mesoendemic setting). Solid lines are the mean of 1000 model runs; shaded areas are the 95% uncertainty intervals (2.5th–97.5th quantiles of stochastic simulations); black circles are the prevalence estimates from the data (Gami[43], Galadimawa[34,44]) with Clopper–Pearson 95% confidence intervals[35]. The pink and blue lines and shaded areas for Galadimawa correspond to $\rho$ values of [(1)]0.65[46] and [(2)]0.3[45], respectively (Table 3).

somatic antigens and those derived from their *Wolbachia* bacterial endosymbionts[18,19]). A more mechanistic approach would be to model (cutaneous and ocular) tissue damage resulting from microfilarial natural death (in generalised onchocerciasis[19]), which could completely or partially resolve over time, but with reduced capacity to heal with cumulative exposure to antigenic stimuli and immunopathological responses. ONCHOSIM assumes that clinical manifestations appear when an individual passes a threshold of accumulated tissue damage which, for irreversible conditions, is permanent, whereas for reversible conditions, resolves once accumulated tissue damage decreases below such a threshold[33]. Supplementary Text 6 and Supplementary Table 3 compare structural and parametric assumptions in morbidity modelling approaches between EPIONCHO-IBM and ONCHOSIM.

Severe itch is thought to be associated with the presence rather than the intensity of infection[26,55]. In our previous modelling approach[22], we had related the prevalence of severe itch to that of adult female worms, following the notion that, under ivermectin MDA, the reduction in prevalence of itch would be smaller and more protracted than that of microfilaridermia[56,57]. Considering the programmatic features in each of the settings used to model the effect of MDA on OSD[39], the projected (relative to baseline) drop in microfilarial prevalence ranged from 8 to 31% after one year of annual ivermectin treatment, and from 22 to 67% 5–6 years into the programme. Model outputs were generally in agreement with the declines in microfilarial

prevalence, except for those settings in which infection prevalence had hardly changed between baseline and impact evaluations. Discrepancies between prevalence estimates and model predictions could be explained by the fact that the data were from a cross-sectional (not a longitudinal) study[39], and microfilarial prevalence was derived from nodule prevalence[41,42]. Our modelled relative reductions in the prevalence of severe itch and RSD ranged, respectively, from 22 to 67% (45% on average) and from 14 to 46% (30% on average) 12 months after the first treatment. These projections compare well with those of a multi-country (Ghana, Nigeria, Uganda) longitudinal follow-up ivermectin trial, in which the prevalence of severe itch decreased by nearly 50% and that of RSD by roughly 33% 12 months after treatment; however, in contrast with our simulations, only individuals aged ≥20 years were recruited into the trial[57].

Our projected baseline prevalence of severe itch (among those aged ≥5 years) for each baseline endemicity level also compares well with modelling results obtained using ONCHOSIM[33]. In the ONCHOSIM study, for meso-, hyper- and highly hyperendemic scenarios, the modelled baseline prevalence of severe itch was approximately 7, 17 and 25%, respectively. According to EPIONCHO-IBM, for hypo- to mesoendemic areas (such as Taraba and Cross River in Nigeria, with 33–50% baseline microfilarial prevalence), the simulated prevalence of severe itch was 9–15%; in hyperendemic areas (Kumba and Ngambe in Cameroon, and Kogi in Nigeria, with 61–63% microfilarial prevalence), the modelled prevalence of severe itch was

18–19%, and for highly hyperendemic areas (Raja in Sudan, with 73% microfilarial prevalence), the projected prevalence of severe itch was 24%.

In contrast, the modelled prevalence of RSD in meso- to highly hyperendemic settings (ranging from 5 to 8%) was substantially lower than that generated by ONCHOSIM (5 to 24%, roughly the same as for severe itch)[33], and also tended to underestimate observations under MDA, except for Taraba. In fact, our estimated probability of developing RSD is about a quarter of the estimated probability of developing severe itch (Table 1), derived from the Kaduna data[26]. ONCHOSIM's simulations used data from a later multi-centre study (in Ghana, Cameroon, Nigeria, Tanzania and Uganda) on 1451 individuals from hypoendemic communities (2–9% nodule prevalence) and 5459 from communities with a wide range of endemicities (6–77% nodule prevalence) aged ≥5 years[55]. In this study, the overall prevalence of severe itch was 32%, nearly twice as high as that of RSD (17%)[55].

The prevalence of microfilaridermia and acute clinical OSD manifestations (severe itch and RSD) decreased markedly during the 20 years of simulated MDA and approached zero in those sites with lower initial endemicity (33–50% microfilarial prevalence) and higher (75–82%) total population coverage despite substantial $\rho$ values (0.8–1.0) in contrast to those with higher endemicity (60–73% infection prevalence) and lower (48–62%) coverage but smaller $\rho$ values (0.1–0.4).

Regarding the irreversible OSD sequelae, EPIONCHO-IBM-modelled baseline prevalence of ATR ranged from 1.3 to 3.6%, of DPM from 1.9 to 5.2%, and of HG from 0.6 to 1.6%, in broad agreement with ONCHOSIM[33], although the latter distinguished between mild (1.6–4.2%) and severe (1.5–9%) depigmentation. A 4.3% DPM prevalence and 1.6% HG prevalence (1128 individuals aged ≥5 years) were reported in a holoendemic savannah area of northern Cameroon[52].

Concerning baseline blindness, our model captured the 10% prevalence in holoendemic Gami[43] and the 5% prevalence in mesoendemic Galadimawa[44]. These values compare well with a predicted 12% in the highly hyperendemic ONCHOSIM scenario and 5% in the mesoendemic scenario[33] (both for savannah onchocerciasis, as in our simulations). For Gami, our modelled reduction of blindness prevalence from baseline was 10%, compared to the recorded 45% decrease after 5 years of MDA[43]. For Galadimawa, our modelled reduction was 20% compared to the reported 79% after 25 years of CDTI[44]. For a highly hyperendemic scenario, ONCHOSIM predicts a blindness prevalence reduction of 25–30% after 5 years of MDA and, for a mesoendemic scenario, a reduction of 80% after 25 years (60–80% coverage)[33], closer to the observed reductions than our predictions. This is likely due to EPIONCHO-IBM lacking excess mortality associated with infection load (our determinant of the probability of developing blindness). ONCHOSIM models excess mortality due to blindness by reducing individuals' remaining life expectancy by a given proportion once they become blind. In turn, excess mortality due to blindness affects the presence of other (non-ocular) clinical manifestations[33]. We have reported a statistically significant density-dependent relationship between microfilarial load and relative risk of mortality[58,59].

Our modelled baseline prevalence of OSD and OOD sequelae for increasing endemicity levels is reasonably consistent with those of ONCHOSIM, with the exception of RSD, the prevalence of which was lower in the Kaduna data[26] we used to parameterise the model compared to the multi-centre study[55] used by ONCHOSIM[33]. As the two models were fitted using different datasets collected at different times (1988–1989[26] for EPIONCHO-IBM; 1994[55] for ONCHOSIM), potential discrepancies in the way criteria were applied to diagnose OSD sequelae[13], and true underlying differences in the prevalence of skin conditions cannot be discounted. Furthermore, the different studies applied different methodologies for assessing severe itch, either by direct questioning[26], or by enquiring about this condition as part of a broader health status survey[39,55]. ONCHOSIM also underestimated the prevalence of ATR and HG compared to the Kaduna data[26], particularly for older age groups[33]. Regarding OOD, we also acknowledge differences in the methodologies used for measuring blindness[28,36,38,43,44], which have been reviewed[60].

A more relevant metric for measuring the impact of MDA on irreversible sequelae such as blindness, would be to consider incidence[28]. Our OAE model indicated a large reduction in OAE incidence under ivermectin MDA, relative to a more modest reduction in OAE prevalence[25]. Both blindness[36] and OAE have been associated with excess mortality[61], another important reason to include excess mortality in EPIONCHO-IBM through its relationship with *O. volvulus* microfilarial load[58,59], as considered in our previous (deterministic) modelling analysis[22]. Currently, for OOD, our model underestimates observed reductions in blindness prevalence. A key priority will, therefore, be to incorporate excess human mortality[58,59], which may more rapidly reduce incident blindness cases[22].

In summary, we have incorporated OSD and OOD into EPIONCHO-IBM, reasonably reproducing their age-prevalence profiles at baseline and modelling the effect of ivermectin MDA on their prevalence trends. Further work will be necessary to improve the modelling of the mechanistic processes that lead to onchocerciasis-associated morbidity, and to investigate the effectiveness and cost-effectiveness, in reducing disease burden, of alternative treatments (e.g. moxidectin[62,63]). Serial cross-sectional studies measuring changes in morbidity prevalence throughout interventions, and where possible, post-intervention, is a critical research gap[64]. Longitudinal studies measuring changes in morbidity prevalence and incidence as a result of interventions would also greatly assist the validation of onchocerciasis morbidity models[65]. Transmission dynamics modelling approaches such as the one presented here would allow for a more nuanced estimation of onchocerciasis-associated morbidity and the impact of MDA, which will be crucial to enhance global burden of disease estimates for onchocerciasis[11]. While modelled morbidity estimates can only provide a simplified representation of the experience of persons living with onchocerciasis, accurately accounting of burden will be crucial to inform progress beyond the 2030 goals[8]. This will be a fruitful area of collaboration between epidemiologists, clinicians and health metrics scientists.

## Data availability

All the data used are available in the publications cited[26,28,29,34,36–39,43–46] and in the Tables and Figures presented. All prevalence estimates, their 95% confidence intervals and model simulation outputs to generate Figs. 3–9 are available in the following Zenodo repository: https://doi.org/10.5281/zenodo.18378708[66].

## Code availability

The model can be found at https://github.com/mrc-ide/EPIONCHO.IBM (https://doi.org/10.5281/zenodo.18401653)[67], for an R code version and https://github.com/adiramani/P_EPIONCHO-IBM (https://doi.org/10.5281/zenodo.18695926)[68] for a Python code version. To enable the reader to reproduce the simulations presented in this paper, see the following vignette file in the above DOI for the R version of the model (Running_EPIONCHO_IBM_with_morbidity.Rmd), explaining how to run the OSD and OOD morbidity models. The code for converting nodule prevalence into microfilarial prevalence is available at https://doi.org/10.5281/zenodo.13969100[42]. Code for reproducing Figs. 3–9 can be found in the Zenodo repository (https://doi.org/10.5281/zenodo.18378708)[66].

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

## Acknowledgements

M.A.D., J.N.S. and M.G.B. acknowledge funding from the MRC Centre for Global Infectious Disease Analysis (grant no. MR/X020258/1), funded by the UK Medical Research Council (MRC). This UK-funded award is carried out in the frame of the Global Health EDCTP3 Joint Undertaking. M.A.D., A.R., M.W. and M.-G.B. also acknowledge funding by the Bill & Melinda Gates Foundation (now Gates Foundation) through the NTD Modelling Consortium (grant no. INV-030046). J.N.S. was funded by the European and Developing Countries Clinical Trials Partnership (EDCTP2, grant no. RIA2017NCT-1843). J.F.M. thanks the Gates Foundation (grant no. OPP1152504). We are grateful to Associate Prof. Luc E. Coffeng for his assistance in the conversion of onchocercal nodule prevalence into microfilarial prevalence, and to Dr Matthew Graham for providing code to calculate the proportion of never-treated population, both of which were necessary to model the impact of ivermectin treatment on onchocerciasis skin disease. We also thank Prof Simon Cousens for discussions during the early stages of this work.

## Author contributions

Conceptualisation: M.A.D., A.R., M.W. and M.-G.B. Data curation: M.A.D. and M.-G.B. Formal analysis, Investigation and Methodology: M.A.D., A.R., M.W. and M.-G.B. Resources: M.E.M, I.E.M and G.A.Z. Software: M.A.D. and A.R. Validation: M.A.D., A.R., M.W., M.E.M., I.E.M. and M.-G.B. Visualisation: M.A.D., A.R. and M.-G.B. Supervision and Funding acquisition: M.W. and M.-G.B. Writing—original draft: M.A.D. and M.-G.B. Writing—review and editing: M.A.D., A.R., M.W., J.N.S., M.E.M, I.E.M., G.A.Z., J.F.M. and M.-G.B.

## Competing interests

The authors declare no competing interests.
