## [Transparent Peer Review file · Communications Medicine]

Modelling of onchocerciasis-associated skin and ocular disease and the impact of ivermectin treatment

Corresponding Author: Dr Matthew Dixon

Version 0:

Reviewer comments:

Reviewer #1

(Remarks to the Author)

This manuscript presents a modelling study that integrates both skin and ocular morbidities into EPIONCHO-IBM to assess the impact of ivermectin mass drug administration (MDA) on onchocerciasis-associated disease burden. The study is methodologically rigorous and draws from historical and contemporary datasets to calibrate model predictions. It represents a significant contribution to the literature and supports policy-oriented modelling aligned with PRIME-NTD principles. However, several important clarifications and updates are needed to strengthen the manuscript, particularly regarding the epidemiological relevance of the estimates, assumptions behind morbidity modelling, and presentation of findings.

Major comments:

1. Use of Outdated Epidemiological Estimates: The manuscript relies heavily on historical datasets (e.g. the 1988–1989 Kaduna survey and early Onchocerciasis Control Programme records) to estimate baseline prevalence and calibrate probabilities for onchocerciasis-associated sequelae. While the authors briefly acknowledge this limitation, the reliance on pre-MDA data to parameterize morbidity risk remains problematic. For instance, Table 1 presents static daily probabilities for developing severe itch, RSD, and irreversible skin conditions, all derived from mf-positive individuals before the scale-up of ivermectin distribution. Rather than attempting to replace these data (given the scarcity of recent, comparable morbidity surveys), the authors should explicitly discuss how the use of these older datasets—collected before widespread MDA—may affect the applicability of current model outputs. They should also reflect on whether the probabilities of sequelae may have changed over time due to (a) reduced cumulative exposure under long-term MDA, (b) improved skin and eye care, or (c) changing environmental or demographic factors that influence exposure or susceptibility. Therefore, highlighting the absence of contemporary data and the need for updated surveys to validate and refine these probabilities would strengthen the manuscript and underscore a broader research gap.

2. Underdetection of Certain Morbidities in Modelling: The model tends to underestimate the prevalence of irreversible onchocerciasis skin disease (OSD) sequelae—such as atrophy, depigmentation, and hanging groin—particularly in older age groups (as seen in Fig. 3D–F). This may reflect the limitation of assigning disease risk based solely on microfilarial positivity rather than microfilarial intensity. While the authors acknowledge this issue in the Discussion (pg. 28–29), the potential for improved model fidelity by incorporating microfilarial load as a risk driver for irreversible sequelae deserves stronger emphasis. The authors should explore this hypothesis more proactively—either through sensitivity analysis or discussion of how a load-based probability model might better reflect cumulative tissue damage and the age gradients observed in real-world data.

Extended explanation to this comment: Currently, the model uses fixed daily probabilities of developing skin disease, applied uniformly to mf-positive individuals (see Table 1: e.g., RSD = 0.042/day, severe itch = 0.164/day). There's no modulation based on how high the microfilarial density is. Since older individuals tend to have longer exposure, and possibly higher historical infection loads, they might accumulate higher risk of irreversible damage. A flat, status-only model (mf-positive = yes/no) does not capture this gradient. That could explain why the model underpredicts irreversible OSD in older age groups.

This was particularly evident in Fig. 3 panels D–F, where simulated age-prevalence curves for atrophy, depigmentation, and hanging groin fall below observed data in the 40–60+ age groups.

3. Comparative Performance vs. ONCHOSIM: While the discussion includes comparisons with ONCHOSIM projections,

these could be better structured. It is unclear how differences in assumed pathophysiology (e.g., threshold models for irreversible conditions in ONCHOSIM) affect projected prevalence under MDA. Consider a table or paragraph explicitly comparing EPIONCHO-IBM and ONCHOSIM assumptions on morbidity progression, recovery, and mortality effects. This would help readers better interpret model discrepancies.

4. Underestimated Impact of Ivermectin on OOD: The model underestimates the observed reduction in blindness (e.g., 20% modelled vs. 79% observed in Galadimawa). This is partly attributed to the absence of excess mortality in EPIONCHO-IBM. However, this raises concerns about the model's ability to reproduce realistic long-term dynamics. Authors should state more clearly how this limitation affects the reliability of long-term burden projections and whether it should be a priority for future model development.

Minor comments:

- Terminology Clarification: Define “reversible” and “irreversible” sequelae clearly early in the manuscript, even if listed in Table 1.
- Figure legends: Some figures (e.g., Fig. 3 and 4) have y-axes on different scales across panels, which may mislead comparisons. Consider harmonizing or flagging more clearly.
- Code Availability: It would be helpful to briefly state in the Methods how readers can reproduce the simulations, not just provide GitHub links at the end.
- Discussion (pg. 28–29): The mention of Sowda and LOD could benefit from a short definition or citation describing this hyper-reactive phenotype for unfamiliar readers.

Reviewer #2

(Remarks to the Author)

Thank you for the opportunity to review this interesting manuscript.

This is an important study to improve on the estimation of the burden of disease and impact of interventions to prevent onchocerciasis-associated skin and ocular morbidity. “Essentially, all models are wrong, but some are useful.” (George Edward Pelham Box). This model adds to the understanding of onchocerciasis associated morbidity and the impact of ivermectin mass drug administration, and paves the way for future modelling to inform the burden of disease and the impact of novel preventive interventions. Onchocerciasis remains a neglected disease of significant public health significance.

The authors describe a sub-model of an existing transmission model of onchocerciasis, EPIONCHO-IBM. They modelled the relationship between onchocerciasis skin disease and prevalence, and between onchocerciasis ocular disease and infection intensity, as well as the impact of ivermectin mass drug administration. The modelled baseline prevalences of age-specific ocular and skin disease were similar to reported estimates, except for irreversible skin disease in older age groups. The model underestimated the impact of ivermectin mass drug administration on the prevalence of blindness and reactive skin disease.

The conclusions are original. The abstract is clear and accessible. The abstract, introduction and conclusions are appropriate.

Limitations include the fact that several factors were not taken into account, including excess mortality associated with infection load, and competing risks, that the model is a simplification of transmission dynamics, assuming that risk is independent of age and that prevalence of skin disease is proportional to incidence or cumulative incidence, and that there are differences in methods to assess skin and ocular disease.

The results are of relevance for a very specialized audience interested in modelling of onchocerciasis and its public health implications.

Statistical tests are appropriate and the description of error bars and probability values are appropriate.

The conclusions and data interpretation are robust and valid, provided the limitations of the model are kept in mind.

The authors suggest several further improvements for future iterations of the model. It is especially important to improve estimation of the reduction of blindness following mass drug administration of ivermectin.

I'm a clinician and epidemiologist, but not an infectious disease modelling specialist. In-depth assessment of methodology of the model is outside of the scope of my expertise and would benefit from additional input from a reviewer with specific technical expertise in infectious disease modelling.

Line by line comments :

78 Probable typo : The 2021, the GBD Study...

166 Is the likelihood of developing reversible OSD equal between people who have previously developed OSD and those who have not ? One would assume that an individual who develops reversible OSD does not return to his baseline risk three days later.

557 Modelled reduction of blindness after how many years ?

Gilles Van Cutsem

Version 1:

Reviewer comments:

Reviewer #1

(Remarks to the Author)

I have reviewed the revised version of the manuscript along with the authors' detailed responses to reviewer comments. I confirm that the authors have adequately addressed all major and minor concerns raised in my initial review.

Notably:

- The justification for using historical datasets for baseline calibration has been strengthened, with added discussion on data limitations, generalizability, and the need for updated morbidity surveys.

- The model's underestimation of irreversible skin disease in older age groups is now better contextualized, and improvements in the revised figures enhance the interpretation of model fit.

- A comparative table outlining differences between EPIONCHO-IBM and ONCHOSIM has been added, improving clarity for readers.

- The underestimation of ivermectin's impact on ocular disease (blindness) is appropriately acknowledged, with proposed refinements identified for future versions of the model.

All minor edits—terminology clarifications, figure annotations, code availability, and explanatory notes—have been implemented appropriately.

The manuscript has been substantially improved following revision and is now suitable for publication. I have no further comments and recommend acceptance.

Reviewer #2

(Remarks to the Author)

I thank the authors for their thorough responses to reviewer's comments. I have no further questions.

made.

Reviewer 1

Reviewer's summary: This manuscript presents a modelling study that integrates both skin and ocular morbidities into EPIONCHO-IBM to assess the impact of ivermectin mass drug administration (MDA) on onchocerciasis-associated disease burden. The study is methodologically rigorous and draws from historical and contemporary datasets to calibrate model predictions. It represents a significant contribution to the literature and supports policy-oriented modelling aligned with PRIME-NTD principles.

However, several important clarifications and updates are needed to strengthen the manuscript, particularly regarding the epidemiological relevance of the estimates, assumptions behind morbidity modelling, and presentation of findings.

Authors' Response. We thank the reviewer for these positive comments, and their recognition of the significant contribution to the literature that our paper can make. We acknowledge the reviewer's comments regarding clarifications/updates and address each in turn below.

Reviewer 1 major comments

Reviewer's comment 1.1. Use of Outdated Epidemiological Estimates: The manuscript relies heavily on historical datasets (e.g. the 1988–1989 Kaduna survey and early Onchocerciasis Control Programme records) to estimate baseline prevalence and calibrate probabilities for onchocerciasis-associated sequelae. While the authors briefly acknowledge this limitation, the reliance on pre-MDA data to parameterize morbidity risk remains problematic. For instance, Table 1 presents static daily probabilities for developing severe itch, RSD, and irreversible skin conditions, all derived from mf-positive individuals before the scale-up of ivermectin distribution.

Rather than attempting to replace these data (given the scarcity of recent, comparable morbidity surveys), the authors should explicitly discuss how the use of these older datasets—collected before widespread MDA—may affect the applicability of current model outputs.

Authors' Response 1.1. Baseline data, collected in the absence of interventions, are essential for calibrating transmission models. Therefore, these datasets are not 'outdated,' but instead very useful, particularly as these baseline surveys were conducted in many locations, prior to the introduction of mass drug administration (MDA) as early as 1988 following the commitment to donate ivermectin by the Mectizan Donation Program (MDP) in 1987. While the clinical examination methods used in the Kaduna (Murdoch *et al.* 2017, Ref [26]) and OCP (Kirkwood *et al.* 1983 [36,37]; Little *et al.* 2004 [28]) datasets have some limitations—which we discuss in the paper—there are no alternative, comparable datasets that precede the introduction of ivermectin MDA. In the case of the OCP datasets, their robustness is supported by the large sample size (12,000 individuals in 53 communities in the case of Kirkwood *et al.* and ~300,000 individuals across 2,315 villages in Little *et al.*). Moreover, as the reviewer notes, there is a scarcity of recent, comparable large-scale surveys, as MDA has commenced in many areas or been implemented for many years. Contemporary pre-control surveys would likely need to focus on onchocerciasis elimination mapping (OEM) areas, which are mostly hypoendemic and, therefore, with low morbidity (the reason they were not prioritised for treatment), making it challenging to estimate reliably the probabilities of developing sequelae. In fact, OEM surveys do not conduct clinical assessments. Clinical assessments for onchocerciasis-associated epilepsy have been conducted in areas with recent or weak control programmes, but in this paper we shift our focus from OAE to OSD and OOD. We have added the following text early in the Discussion section:

“We used extensive baseline data, collected prior to the implementation of interventions, for calibrating the model to pre-control transmission conditions. The parameterisation of the probabilities of developing OSD and OOD sequelae was also derived from pre-intervention datasets. Rather than outdated, such data remain highly relevant, robust, and essential given the lack of recent, large-scale pre-control morbidity surveys.”

We also thank the reviewer for raising the point about applicability. An additional dimension to this is that our parameterisations for ocular disease are based on savannah setting datasets (Kirkwood *et al.*, 1983 [36]; Abiose *et al.*, 1994 [38]; Little *et al.* 2004 [28]), and while the notion of ‘strain’-specific differences in blindness prevalence between savannah and forest onchocerciasis settings has been largely accepted, Cheke *et al.* (new reference [54] in the revised manuscript) have reappraised the situation, indicating an absence of notable differences between ‘savannah and forest strains’ regarding blindness rates. This supports the more general applicability of our EPIONCHO-IBM-derived blindness dynamics. We have added a sentence in the Discussion as follows:

“Our parameterisation of the probability of developing blindness was based on data from savannah settings²⁸. Given that a recent re-appraisal of blindness prevalence in savannah and forest settings reported no appreciable differences due to ‘savannah’ and ‘forest’ parasite ‘strains’⁵⁴, we consider that our results are generalisable.”

New reference [54] is:

Cheke, R. A., Little, K. E., Young, S., Walker, M. & Basáñez, M.-G. Taking the strain out of onchocerciasis? A reanalysis of blindness and transmission data does not support the existence of a savannah blinding strain of onchocerciasis in West Africa. *Adv. Parasitol.* **112**, 1–50 (2021).

Reviewer’s comment 1.2. They should also reflect on whether the probabilities of sequelae may have changed over time due to (a) reduced cumulative exposure under long-term MDA, (b) improved skin and eye care, or (c) changing environmental or demographic factors that influence exposure or susceptibility. Therefore, highlighting the absence of contemporary data and the need for updated surveys to validate and refine these probabilities would strengthen the manuscript and underscore a broader research gap.

Authors’ Response 1.2. This is an important point to clarify, and we welcome the opportunity to do so. Regarding point (a), the simulated implementation of MDA in EPIONCHO-IBM drives dynamic changes in microfilarial prevalence and intensity over time, under the model assumptions of the effects of ivermectin on the parasite and programmatic parameters. Consequently, the modelled prevalences of sequelae decline because of these underlying dynamics, rather than requiring adjustments to the probabilities of developing the sequelae. Our model therefore accounts for decreasing exposure risk due to interventions.

Improved skin and eye care (b) would have to be modelled as a particular intervention based on data for specific settings, but again, would not affect the intrinsic probabilities of developing sequelae as functions of infection.

Concerning the impact of environmental or demographic change (c), relevant evidence of long-term deforestation, land-use trends, demographic transitions, population changes, etc. would be necessary to understand changes in exposure risk. For instance, environmental change is anticipated to influence vector density, distribution and species composition, and

data on these factors, or evidence of change in epidemiological conditions not driven by control efforts would be required to model any secular transmission trend in a particular area (see for instance Post *et al.* 2022 <https://doi.org/10.1371/journal.pntd.0010684>, and Ngave *et al.* (2025) <https://doi.org/10.21203/rs.3.rs-7151190/v1>, who report secular changes likely due to environmental change in the Ituri Province of the DRC).

Regarding the reviewer's suggestion to highlight the need for updated surveys as a research gap, serial cross-sectional studies measuring changes in prevalence throughout interventions, and where possible, post-intervention, is indeed a critical research gap for further validation of onchocerciasis morbidity modelling. For example, Otabil *et al.* (2023) (Ref [16]) found onchocerciasis skin disease (OSD)-related morbidities to be prevalent after 27 years of ivermectin treatment (using the WHO Skin NTD App). More studies of this nature are required. See also Borlase *et al.* 2023 <https://doi.org/10.1098/rstb.2022.02792023>. We have added text and additional references in the final paragraph of the manuscript to address this:

“Serial cross-sectional studies measuring changes in morbidity prevalence throughout interventions, and where possible, post-intervention, is a critical research gap⁶⁴. Longitudinal studies measuring changes in morbidity prevalence and incidence as a result of interventions would also greatly assist the validation of onchocerciasis morbidity models⁶⁵”.

New Ref [64] is:

Borlase, A., Prada, J. M. & Crellen, T. Modelling morbidity for neglected tropical diseases: the long and winding road from cumulative exposure to long-term pathology. *Philos. Trans. R. Soc. Lond. B Biol. Sci.* **378**, 20220279 (2023).

Although new Ref [65]) does not refer specifically to OSD or OOD, it illustrates the importance of conducting morbidity longitudinal studies:

New Ref [65] is:

Jada, S. R. et al. Effect of onchocerciasis elimination measures on the incidence of epilepsy in Maridi, South Sudan: a 3-year longitudinal, prospective, population-based study. *Lancet Infect. Dis.* **11**, e1260–e1268 (2023).

Reviewer's comment 1.3. Under-detection of Certain Morbidities in Modelling: The model tends to underestimate the prevalence of irreversible onchocerciasis skin disease (OSD) sequelae—such as atrophy, depigmentation, and hanging groin—particularly in older age groups (as seen in Fig. 3D–F). This may reflect the limitation of assigning disease risk based solely on microfilarial positivity rather than microfilarial intensity.

While the authors acknowledge this issue in the Discussion (pg. 28–29), the potential for improved model fidelity by incorporating microfilarial load as a risk driver for irreversible sequelae deserves stronger emphasis. The authors should explore this hypothesis more proactively—either through sensitivity analysis or discussion of how a load-based probability model might better reflect cumulative tissue damage and the age gradients observed in real-world data.

Extended explanation to this comment: Currently, the model uses fixed daily probabilities of developing skin disease, applied uniformly to mf-positive individuals (see Table 1: e.g., RSD = 0.042/day, severe itch = 0.164/day). There's no modulation based on how high the microfilarial density is. Since older individuals tend to have longer exposure, and possibly

higher historical infection loads, they might accumulate higher risk of irreversible damage. A flat, status-only model (mf-positive = yes/no) does not capture this gradient. That could explain why the model under-predicts irreversible OSD in older age groups. This was particularly evident in Fig. 3 panels D–F, where simulated age-prevalence curves for atrophy, depigmentation, and hanging groin fall below observed data in the 40–60+ age groups.

Authors' Response 1.3. We thank the reviewer for raising this important point. In fact, we did investigate the relationship between the probability of developing OSD and microfilarial load and had presented these results in Fig. 1 of the Supplementary file in the original submission. With the exception of skin atrophy, these relationships were essentially described by a flat, horizontal curve and therefore deemed uninformative (see Supplementary Fig. 1). To capture the effects that the reviewer indicates, modelling cumulative (rather than current) microfilarial load may be more appropriate. This is certainly an area for further model improvement. We have moved, earlier in the revised Discussion, and included additional text in the paragraph explaining that incorporating an age-specific force-of-infection would improve upon simpler assumptions of constant morbidity risk or linear accumulation of morbidity:

“Our modelling approach for skin disease linked the probability of developing OSD sequelae to infection status (being mf-positive) rather than to infection intensity (microfilarial load), because in the dataset we used²⁶, the latter was not particularly informative. This approach—for both reversible (severe itch; RSD) and irreversible (ATR; DPM; HG) OSD—relies on the broad assumption that prevalence is proportional to incidence (for reversible conditions) or to cumulative incidence (for irreversible conditions). In both cases, this simplification assumes stable transmission dynamics, age-independent risk, and negligible competing risks (e.g., from death or migration)—conditions that may not fully hold, but which nonetheless yield reasonable approximations under endemic equilibrium and limited data. A more comprehensive alternative would involve a dynamic model of morbidity risk, incorporating estimates of age-specific force of infection derived from age-stratified infection prevalence data⁵³. This would avoid the need to assume constant exposure over age, constant morbidity risk, or linear accumulation of morbidity, and would instead allow infection and morbidity to be modelled explicitly as age-structured processes under more flexible assumptions.”

We also take the opportunity to clarify that original Figure 3 (age-prevalence profiles of OSD) in the first submission was based on an incorrect formulation of the probabilities for irreversible OSD. We have now updated Figure 3 to reflect the corrected irreversible OSD probabilities presented in revised Table 1, calculated according to Equation (1). The age-prevalence profile for depigmentation (Fig. 3E) now more closely matches the data. Figure 8 has also been updated.

Reviewer's comment 1.4. Comparative Performance vs. ONCHOSIM: While the discussion includes comparisons with ONCHOSIM projections, these could be better structured. It is unclear how differences in assumed pathophysiology (e.g., threshold models for irreversible conditions in ONCHOSIM) affect projected prevalence under MDA. Consider a table or paragraph explicitly comparing EPIONCHO-IBM and ONCHOSIM assumptions on morbidity progression, recovery, and mortality effects. This would help readers better interpret model discrepancies.

Authors' Response 1.4. We thank the reviewer for this suggestion. We have prepared a comparative table and included OAE in addition to OSD and OOD for the sake of

completeness. We have added the following text in the Discussion section to signpost readers to new Supplementary Text 6 and Supplementary Table 3:

“Supplementary Text 6 and Supplementary Table 3 compare structural and parametric assumptions in morbidity modelling approaches between EPIONCHO-IBM and ONCHOSIM.”

Reviewer’s comment 1.5. Underestimated Impact of Ivermectin on OOD: The model underestimates the observed reduction in blindness (e.g., 20% modelled vs. 79% observed in Galadimawa). This is partly attributed to the absence of excess mortality in EPIONCHO-IBM. However, this raises concerns about the model's ability to reproduce realistic long-term dynamics. Authors should state more clearly how this limitation affects the reliability of long-term burden projections and whether it should be a priority for future model development.

Authors’ Response 1.5. We appreciate the reviewers’ feedback and recognise that our model currently underestimates the impact of ivermectin MDA on OOD, which is a priority area for future model refinement. This is likely owing to the absence of excess mortality associated with microfilarial load in our model, which we had already reported (Refs [58] and [59]). We have acknowledged this in the Discussion. Modelling incidence data will also be important, as we have previously done for OAE. We have included the following text in the Discussion:

“Currently, for OOD, our model underestimates observed reductions in blindness prevalence. A key priority will, therefore, be to incorporate excess human mortality^{58,59}, which may more rapidly reduce incident blindness cases²².”

Minor comments

Reviewer’s comment 1.6. Terminology Clarification: Define “reversible” and “irreversible” sequelae clearly early in the manuscript, even if listed in Table 1.

Authors’ Response 1.6. We are grateful to the referee for this useful suggestion which will be helpful to the readers. We have included a sentence early in the Methods section, under Onchocerciasis skin disease (OSD) to include these definitions:

“We differentiate OSD sequelae into reversible (severe itch and RSD), where an individual can revert to being sequela-negative, and irreversible (ATR, DPM, HG), where an individual remains sequela-positive for the remainder of their life.”

Reviewer’s comment 1.7. Figure legends: Some figures (e.g., Fig. 3 and 4) have y-axes on different scales across panels, which may mislead comparisons. Consider harmonizing or flagging more clearly.

Authors’ Response 1.7. Given the differences in prevalence magnitudes across sequelae, presenting a consistent y-axis across all plots would lead to several age-prevalence curves becoming very compressed and difficult to visualise. This would be a particular issue for Figure 3, panels D-F. In Figure 4, y-axes are already consistent across panels. To highlight the difference in y-axes in Figure 3, we had already included text to bring the readers’ attention to this in the original submission: “NB: y-axis in different scales”. (NB stands for ‘nota bene’.)

Reviewer's comment 1.8. Code Availability: It would be helpful to briefly state in the Methods how readers can reproduce the simulations, not just provide GitHub links at the end.

Authors' Response 1.8. We thank the reviewer for this suggestion. Rather than adding additional text to the Methods, we have expanded the Code Availability statement to include a link to the GitHub vignette that provides a full model description and step-by-step instructions for using the model, enabling readers to reproduce the simulations. The additional text in the 'Code availability' statement reads as follows:

"To enable the reader to reproduce the simulations presented in this paper, see the following vignette on how to run the OSD and OOD morbidity models in EPIONCHO-IBM: [https://github.com/mrc-ide/EPIONCHO.IBM/blob/master/vignettes/RunningEPIONCHO IBM with morbidity.Rmd](https://github.com/mrc-ide/EPIONCHO.IBM/blob/master/vignettes/RunningEPIONCHO%20IBM%20with%20morbidity.Rmd)"

Reviewer's comment 1.9. Discussion (pg. 28–29): The mention of Sowda and LOD could benefit from a short definition or citation describing this hyper-reactive phenotype for unfamiliar readers.

Authors' Response 1.9. We agree that this would help the readers to better understand LOD and 'Sowda'. We have included the following text:

"LOD is characterised by pruritic, hyperpigmented, hyperkeratotic plaques, usually in an asymmetrical distribution involving one limb (also known as 'sowda')¹³. LOD results from heightened host immunological responses which kill microfilariae at the expense of substantial skin pathology¹⁷."

References [13] and [17] had been included in the original submission.

Reviewer 2

Reviewer's summary: Thank you for the opportunity to review this interesting manuscript. This is an important study to improve on the estimation of the burden of disease and impact of interventions to prevent onchocerciasis-associated skin and ocular morbidity. "Essentially, all models are wrong, but some are useful." (George Edward Pelham Box). This model adds to the understanding of onchocerciasis associated morbidity and the impact of ivermectin mass drug administration, and paves the way for future modelling to inform the burden of disease and the impact of novel preventive interventions. Onchocerciasis remains a neglected disease of significant public health significance.

The authors describe a sub-model of an existing transmission model of onchocerciasis, EPIONCHO-IBM. They modelled the relationship between onchocerciasis skin disease and prevalence, and between onchocerciasis ocular disease and infection intensity, as well as the impact of ivermectin mass drug administration. The modelled baseline prevalences of age-specific ocular and skin disease were similar to reported estimates, except for irreversible skin disease in older age groups. The model underestimated the impact of ivermectin mass drug administration on the prevalence of blindness and reactive skin disease. The conclusions are original. The abstract is clear and accessible. The abstract, introduction and conclusions are appropriate.

Authors' Response. We thank the reviewer for the overall summary of our work, the positive comments and finding value in our work.

Reviewer's comment 2.1. Limitations include the fact that several factors were not taken into account, including excess mortality associated with infection load, and competing risks, that the model is a simplification of transmission dynamics, assuming that risk is independent of age and that prevalence of skin disease is proportional to incidence or cumulative incidence, and that there are differences in methods to assess skin and ocular disease.

Authors' Response 2.1. The reviewers' appraisal of our limitations is correct, identifying the areas we have focussed on as future research avenues (e.g., age-specific force-of-infection and risk, incorporating excess-mortality) or research gaps (e.g., need for longitudinal studies) (see **Authors' Response 1.2, 1.3, and 1.5**). Please note that in response to feedback received from the Editorial team, we have now integrated the "Limitations and future model refinements" text into the discussion, rather than presenting them as a separate section.

Reviewer's comment 2.2. The results are of relevance for a very specialized audience interested in modelling of onchocerciasis and its public health implications.

Authors' Response 2.2. While we agree that the manuscript is particularly relevant for a readership involved in epidemiology, control and modelling of onchocerciasis, we contend that the quantification of morbidity in relation to infection has broader relevance across the field of parasitic infections in general and neglected tropical diseases in particular. For example, Borlase *et al.* (2023) highlight this as a critical research gap for NTDs. More generally, the use of transmission dynamics modelling to quantify disease burden is of wide interest in infectious disease research, particularly in the context of enhancing the Global Burden of Disease study.

Borlase, A., Prada, J. M. & Crellen, T. Modelling morbidity for neglected tropical diseases: the long and winding road from cumulative exposure to long-term pathology. Philos. Trans. R. Soc. Lond. B Biol. Sci. 378, 20220279 (2023). <https://doi.org/10.1098/rstb.2022.0279>.

Reviewer's comment 2.3. Statistical tests are appropriate and the description of error bars and probability values are appropriate. The conclusions and data interpretation are robust and valid, provided the limitations of the model are kept in mind.

Authors' Response 2.3. We thank the referee for these comments.

Reviewer's comment 2.4. The authors suggest several further improvements for future iterations of the model. It is especially important to improve estimation of the reduction of blindness following mass drug administration of ivermectin.

Authors' Response 2.4. We agree that this is a particular area of model development and improvement. Please see **Authors' Response 1.5**. We have included the following text in the Discussion:

"Currently, for OOD, our model underestimates observed reductions in blindness prevalence. A key priority will, therefore, be to incorporate excess human mortality^{58,59} and model blindness incidence with EPIONCHO-IBM (as previously done with a deterministic precursor²²)."

Reviewer’s comment 2.5. I’m a clinician and epidemiologist, but not an infectious disease modelling specialist. In-depth assessment of methodology of the model is outside of the scope of my expertise and would benefit from additional input from a reviewer with specific technical expertise in infectious disease modelling.

Authors’ Response 2.5. Reviewer 1 has indicated expertise in modelling; therefore, for specific technical comments and responses in relation to the transmission and morbidity modelling, please see our responses to Reviewer 1.

Reviewer’s comment 2.6. 78 Probable typo: The 2021, the GBD Study...

Authors’ Response 2.6. We thank the reviewer for spotting this and have corrected to:

“The 2021 GBD Study ...”

Reviewer’s comment 2.7. 166 Is the likelihood of developing reversible OSD equal between people who have previously developed OSD and those who have not? One would assume that an individual who develops reversible OSD does not return to his baseline risk three days later.

Authors’ Response 2.7. We thank the reviewer for this insightful point. Modelling reversible OSD risk conditional on previous morbidity is challenging, as it would require detailed data on the probability of developing reversible OSD given past episodes. ONCHOSIM addresses this by modelling OSD dynamics based on cumulative tissue damage. This approach could be explored as a future refinement of EPIONCHO-IBM. As a result of a request by Reviewer 1, we have included a detailed comparison of the morbidity modelling approaches by the two models (see **Authors’ Response 1.4**). We acknowledge the point made by the reviewer and have added the following text in the revised Discussion, in the paragraph describing the duration of reversible OSD sequelae and the current lack of understanding of their natural histories:

“Also, the probability of developing reversible OSD may not be the same between individuals who have previously developed it and those who develop it for the first time”.

Reviewer’s comment 2.8. 557 Modelled reduction of blindness after how many years?

Authors’ Response 2.8. We already include reference to this, stating a “*decrease after 5 years of MDA*”. We have highlighted this in the revised version of the manuscript to facilitate inspection by the referee.